# Effects of ozone-vegetation interactions on meteorology and air quality in China using a two-way coupled land-atmosphere model

Jiachen Zhu[1], Amos P.K. Tai[2], Steve Hung Lam Yim[3,4]

[1]Department of Geography and Resource Management, The Chinese University of Hong Kong, Sha Tin, N.T., Hong Kong, China

[2]Earth System Science Programme, The Chinese University of Hong Kong, Sha Tin, N.T., Hong Kong, China

[3]Asian School of the Environment, Nanyang Technological University, 50 Nanyang Avenue, 639798 Singapore

[4]Lee Kong Chian School of Medicine, Nanyang Technological University, Singapore

*Correspondence to*: Steve H.L. Yim (yimsteve@gmail.com) & Amos P.K. Tai (amostai@cuhk.edu.hk)

**Abstract.** Tropospheric ozone ($O_3$) is one of the most important air pollutants in China and is projected to continue to increase in the near future. $O_3$ and vegetation closely interact with each other and such interactions may not only affect plant physiology (e.g., stomatal conductance and photosynthesis) but also influence the overlying meteorology and air quality through modifying leaf stomatal behaviors. Previous studies have highlighted China as a hotspot in terms of $O_3$ pollution and $O_3$ damage to vegetation. Yet, few studies have investigated the effects of $O_3$-vegetation interactions on meteorology and air quality in China, especially in the light of recent severe $O_3$ pollution. In this study, a two-way coupled land-atmosphere model was applied to simulate $O_3$ damage to vegetation and the subsequent effects on meteorology and air quality in China. Our results reveal that $O_3$ causes up to 16% enhancement in stomatal resistance, whereby large increases are found in Henan, Hebei and Shandong provinces. $O_3$ damage causes more than 0.6 $\mu$mol $CO_2$ m$^{-2}$ s$^{-1}$ reductions in photosynthesis rate, and at least 0.4 and 0.8 g C m$^{-2}$ day$^{-1}$ decrease in leaf area index (LAI) and gross primary production (GPP), respectively, and hotspot areas appear in the northeastern and southern China. The associated reduction in transpiration causes a 5–30 W m$^{-2}$ decrease (increase) in latent heat (sensible heat) flux, which induces a 3% reduction in surface relative humidity, 0.2–0.8 K increase in surface air temperature, and 40–120 m increase in boundary layer height in China. We also found that the meteorological changes further induce a 2–6 ppb increase in $O_3$ concentration in northern and south-central China mainly due to enhanced isoprene emission following increased air temperature, demonstrating that $O_3$-vegetation interactions can lead to strong positive feedback that can amplify $O_3$ pollution in China. Our findings emphasize the importance of considering the effects of $O_3$ damage and $O_3$-vegetation interactions in air quality simulations, with ramifications for both air quality and forest management.

## 1. Introduction

Tropospheric ozone ($O_3$) is a secondary air pollutant, which is mainly formed from the photochemical oxidation of carbon monoxide (CO), methane ($CH_4$) and non-methane volatile organic compounds (VOCs) by hydroxyl radicals (OH) in the presence of nitrogen oxides ($NO_x = NO + NO_2$). $O_3$ is known as the third most important greenhouse gas with an estimated radiative forcing of 0.41 W m$^{-2}$ for the period of 1750–2010 (IPCC, 2013; Stevenson et al., 2013). As an air pollutant, $O_3$ is also shown to be harmful to not only human health but also vegetation and crop health (Anenberg et al., 2010; Cohen et al., 2017). Various field experiments and numerical modeling studies have already demonstrated that $O_3$ can not only reduce gross primary production (GPP) of natural vegetation as well as crop yields (Ainsworth et al., 2012; Lombardozzi et al., 2012; Tai e al., 2014; Feng et al., 2015; Yue et al., 2017; Li et al., 2018), but also decrease transpiration (Arnold et al., 2018), decrease runoff (Li et al., 2016) on larger scales and therefore affect the global carbon and water cycle (Lombardozzi et al., 2015).

Vegetation can in turn modulate $O_3$ concentration through influencing the sources and sinks of $O_3$. Dry deposition of $O_3$ onto vegetation is a major sink for $O_3$, mainly via stomatal uptake. Stomata are the pores on plant leaves; they control water exiting and carbon entering the leaf interior and hence influence the water and carbon exchange between the land and atmosphere. When vegetation is exposed to enhanced $O_3$ levels, cellular and tissue damage can result in a decrease in photosynthesis rate, thus altering $CO_2$ assimilation. Stomata conductance may decrease subsequently in response to $O_3$ exposure, thus reducing the dry-depositional sink of $O_3$ (Sadiq et al., 2017; Zhou et al., 2018), but some studies also suggest that $O_3$ exposure can cause stomata to respond more sluggishly to changing environmental conditions, such as drought, with complex overall effects on stomatal behaviors and dry deposition (e.g., Huntingford et al., 2018). Moreover, recent studies showed reduced dry deposition velocities of $O_3$ by drought-stressed vegetation, which affects surface $O_3$ trends and extremes (Huang et al., 2016; Lin et al., 2019; Lin et al., 2020). Vegetation also affects the sources of $O_3$; the most abundant biogenic VOC (BVOC) species emitted by vegetation is isoprene ($C_5H_8$), which is a major precursor for $O_3$ formation in polluted, high-$NO_x$ environments, but removes $O_3$ by ozonolysis or by sequestering $NO_x$ in more pristine, low-$NO_x$ regions (Hollaway et al., 2017). Isoprene production is known to be highly coupled with photosynthesis and by extension to stomatal conductance (Arneth et al., 2007). Moreover, transpiration, which is modulated by stomatal behaviors, significantly regulates surface meteorology including water vapor content and air temperature, which further influence the production and loss of $O_3$. Therefore, through influencing plant ecophysiology (e.g., photosynthesis and stomata behaviors), $O_3$-vegetation interactions can modulate boundary-layer meteorology, climate, and may further affect $O_3$ air quality via a series of feedback mechanisms. It is therefore essential to fully understand the $O_3$-vegetation interactions and the following climatic and biospheric impacts especially in areas with high $O_3$ concentrations and vegetation density.

In many land surface and biospheric models, such as Noah-Multi Parameterization (Noah-MP) or Community Land Model (CLM), the Farquhar-Ball-Berry model (FBB, Farquhar et al., 1980; Ball et al., 1987) is commonly used to simulate stomatal conductance and photosynthetic rate. In the FBB model, the calculation of stomata conductance is based on the calculation of photosynthesis, which makes them tightly coupled with each other. Therefore, in several land surface models that consider $O_3$ damage effect on vegetation, the photosynthetic rate is modified first and the stomatal conductance is modified subsequently, which means stomata conductance and photosynthesis will change collinearly under

chronic $O_3$ exposure (Sitch et al., 2007; Yue and Unger, 2014). However, field experiments have shown that, under chronic $O_3$ exposure, stomata conductance decreases with a smaller magnitude than photosynthetic rate does, which makes the simulations of stomata conductance and photosynthetic rate as well as the following water and carbon cycles in the above models less accurate (Lombardozzi et al., 2012). Modifying stomata conductance and photosynthesis separately in land surface models is therefore more reasonable. Lombardozzi et al (2012) modified the stomata conductance and photosynthetic rate separately based on the cumulative uptake of $O_3$ into leaves and has shown a better representation of plant responses to $O_3$ exposure. Efforts have been made to investigate the effects of $O_3$ exposure on land biosphere based on the above $O_3$ damage schemes. For example, based on an off-line process-based vegetation model, Yue and Unger (2014) found that $O_3$ damage decrease GPP by 4–8% on average in the eastern US and leads to significant decreases of 11–17% in east coast hot spots. Using the offline CLM model, Lombardozzi et al. (2015) estimated that the present $O_3$ exposure reduces GPP and transpiration globally by 8–12% and 2.0–2.4%, respectively.

Several modeling studies conducted so far have demonstrated the importance of considering the interactions and feedbacks between atmosphere and biosphere. By dynamically coupling $O_3$ and LAI but without considering the meteorological feedbacks of $O_3$-vegetation interactions to $O_3$, Zhou et al. (2018) found that $O_3$-induced damage on LAI can lead to changes in $O_3$ concentrations by −1.8 to +3 ppb in boreal summer. By considering the interactions between atmospheric chemistry with biosphere in a two-way coupling model, Lei et al. (2020) quantified the damaging effects of $O_3$ on vegetation and found a global reduction of annual GPP by 1.5–3.6 %, with regional extremes of 10.9–14.1 % in the eastern US and eastern China. Based on the CESM model with fully interactive atmospheric chemistry, biogeochemical and biogeophysical cycles, Sadiq et al. (2017) estimated that surface $O_3$ is 4–6 ppb higher in Europe, North America and China in simulations with $O_3$-vegetation coupling comparing the surface $O_3$ concentrations without $O_3$-vegetation coupling. Based on modified WRF-Chem model, Li et al (2016, 2018) investigated the effect of $O_3$ exposure on hydroclimate and crop productivity in the US, and highlighted $O_3$ damage effects on meteorological fields and surface energy balance as well as the crop yields, but the feedbacks of changing meteorology onto surface $O_3$ were not investigated. Arnold et al (2018) examined the global climate response to $O_3$ exposure and found $O_3$ damage on vegetation can induce widespread surface warming and changes in clouds, which could be critical on regional scales. Although the interactions between $O_3$ and vegetation are critical to our environment, adequate representation of $O_3$-vegetation interactions is still missing in most atmospheric models used for climate and atmospheric chemistry simulations, at least in part due to incomplete coupling capacities with land surface or biospheric model components at high resolutions, and in part due to limited observations to optimize $O_3$ damage schemes for wider regional applicability.

With the rapid urbanization and industrialization in the recent decades, China has experienced increasingly severe $O_3$ pollution, which is expected to continue to worsen in the near future. $O_3$ concentration in China has been observed to exceed ambient air quality standard by 100–200% (Wang et al., 2017) with the maximum 8-hour mean concentration of $O_3$ (MDA8 $O_3$) increasing by 4.6% per year from 2015 to 2017 (Silver et al., 2018). Lu et al. (2018) showed that urban surface $O_3$ in China during 2013–2017 was significantly higher than that in other regions around the world, and thus vegetation exposure to $O_3$ is also higher in China. Li et al. (2018) also revealed the increasing trend of $O_3$ in megacity clusters of China during 2013–2017, which is closely related with meteorology,

anthropogenic emissions and $PM_{2.5}$ concentrations. Global-scale studies have highlighted China as a hotspot of $O_3$ pollution and damage to vegetation compared with other regions (Sadiq et al., 2017; Arnold et al., 2018; Lei et al., 2020). However, a comprehensive study of how $O_3$ affects meteorology and air quality through $O_3$-vegetation interactions in China at high spatial resolutions, especially under severe $O_3$ pollution, is still limited but highly needed. Moreover, there have been limited studies focusing on the feedbacks of $O_3$-vegetation coupling on $O_3$ concentration itself, especially in China, which is one of the main scopes of our study.

This study, therefore, first adopted and implemented a semi-mechanistic $O_3$ damage scheme in a widely used regional atmosphere-land modeling framework and hence used it to simulate and assess the impacts of $O_3$-vegetation interactions on boundary-layer meteorology and air quality in China at a high spatial resolution. Specifically, $O_3$-induced damage to vegetation, changes in meteorology in China due to $O_3$-vegetation coupling, and the subsequent feedback effects onto $O_3$ concentration itself are examined, which is crucial to fully understand the $O_3$-vegetation interactions and the following impacts on climate, biosphere, and air quality in areas with both high $O_3$ concentrations and high vegetation coverage.

## 2. Methods
## 2.1 WRF-Chem Model Setup

The Weather Research and Forecasting (WRF) model is a state-of-the-art mesoscale nonhydrostatic meteorological model. An atmospheric chemistry module that includes various gas-phase chemistry and aerosol mechanisms has been implemented into and fully coupled with WRF to create the WRF-Chem model (Grell et al., 2005; Fast et al., 2006). In WRF-Chem, both the air quality and meteorological components use the same transport scheme, model grid, subgrid-scale transport physics and time step. WRF-Chem has been widely used in previous air quality studies (e.g., Li et al., 2016; Li et al., 2018; Liu et al., 2018; Liu et al., 2020). In this study, we applied our revised WRF-Chem model based on version 3.8.1 to simulate meteorological fields and $O_3$ concentration over China. Simulations are conducted from 24 May to 1 September every year from 2014 to 2017 and the days in May were discarded as spin-up. For the land surface component within WRF, we used Noah-MP, which will be described in the next subsection.

The model domain was configured at a horizontal resolution of 27 km on the Lambert Conformal projection, centered at 37°N, 108.1°E and covering the whole China. The model has 26 vertical layers, with the lowest layer at 0.17 km and the highest layer at 17.67 km. The meteorological initial and boundary conditions are provided by the 6-hourly Final Operational Global Analysis (FNL) dataset at a horizontal resolution of 1°×1°. The chemical initial and boundary conditions were generated from the Model for Ozone and Related Chemical Tracer version 4 (MOZART-4), which is available at a horizontal resolution of 1.9°×2.5° with 56 vertical layers (Emmons et al., 2010).

Anthropogenic emissions were from the Multi-resolution Emission Inventory for China (MEIC) compiled at a spatial resolution of 27 km and a 1-hourly temporal resolution suitable for our research domain. Biogenic emissions were calculated online by the Model of Emissions of Gases and Aerosol from Nature (MEGAN) (Guenther et al., 2006). Biomass burning emissions were extracted from the Fire Inventory from NCAR (FINN) version 1.5 datasets (Wiedinmyer et al., 2010). Dust emissions were

generated online by the Goddard Global Ozone Chemistry Aerosol Radiation and Transport model (GOCART; Ginoux et al., 2001). Gas-phase chemistry was simulated with second generation Regional Acid Deposition Model (RADM2; Stockwell et al., 1990) mechanism, and the Modal Aerosol Dynamics Model for Europe (MADE; Ackermann et al., 1998), which is coupled with the Secondary Organic Aerosol Model (SORGAM; Schell et al., 2001) for aerosol treatment. Detailed physics schemes used in the simulations are shown in Table S1.

## 2.2 Description of Noah-MP model

Noah-MP is a land surface model that uses multiple options for key land-atmosphere interaction processes (Niu et al., 2011). Noah-MP contains a separate vegetation canopy defined by a canopy top and bottom, crown radius, and leaves with prescribed dimensions, orientation, density, and radiometric properties. The canopy employs a two-stream radiation transfer approach along with shading effects necessary to achieve proper surface energy and water transfer processes (Dickinson, 1983). Noah-MP is capable of distinguishing between $C_3$ and $C_4$ photosynthesis pathways and defines vegetation-specific parameters for plant photosynthesis and respiration.

Noah-MP is available for prognostic vegetation growth that combines a Ball-Berry photosynthesis-based stomatal resistance (Farquhar et al., 1980; Ball et al., 1987) that allocates carbon to various parts of vegetation (leaf, stem, wood and root) and soil carbon pools (fast and slow). GPP, leaf area index (LAI) and canopy height are then predicted downstream from photosynthesis. Noah-MP also considers the photosynthesis of sunlit and shaded leaves separately, whereby sunlit leaves are more limited by $CO_2$ concentration while shaded leaves are more constrained by insolation, which may thus have different responses to $O_3$ damage. The dynamic LAI and canopy height calculation will further affect surface energy fluxes, which will then affect the boundary-layer meteorology when coupling with the atmosphere model in WRF-Chem. The land use types and the vegetation parameters are based on the U.S. Geological Survey (USGS) embedded in Noah-MP. Fig. 1 shows the spatial distribution of vegetation fraction of dominant vegetation types in China. The distribution of main vegetation groups (broadleaf, needleleaf, crop and grass) that have different sensitivities to $O_3$ damage following Lombardozzi et al. (2015) are shown in Fig. 1.

In this study, the $O_3$ concentration simulated by the chemical module of the WRF-Chem model was also dynamically passed onto the Noah-MP land surface model at every time step to modify the photosynthesis and stomatal conductance due to $O_3$ damage. The land surface variables simulated by Noah-MP were also dynamically passed back onto the atmospheric components, thus allowing immediate, two-way feedback effects onto meteorological fields, $O_3$ and other atmospheric chemical constituents. In this way, land surface processes, atmospheric dynamics, and atmospheric chemistry in the WRF-Chem model were fully coupled.

## 2.3 $O_3$ damage parameterization

In Noah-MP, the Farquhar model (Farquhar et al., 1980) was used to calculate photosynthetic rate, whereas Ball-Berry model was used to calculate stomatal conductance (Ball et al., 1987). The

photosynthesis rate, $A$ (μmol $CO_2$ m$^{-2}$ s$^{-1}$), is calculated separately for sunlit and shaded leaves and is limited by either one of three limiting factors and can be calculated as

$$A = \min(W_c, W_j, W_e) I_{gs} \tag{1}$$

where $W_c$ is the Rubisco-limited photosynthesis rate, $W_j$ is the light-limited photosynthesis rate, and $W_e$ is the export-limited photosynthesis rate. $I_{gs}$ is the growing season index with values ranging from 0 to 1. Stomatal conductance ($g_s$) is computed based on the photosynthesis rate from the Farquhar model as

$$g_s = \frac{1}{r_s} = m\frac{A}{c_s}\frac{e_s}{e_i}P_{atm} + b \tag{2}$$

where $g_s$ is the leaf stomatal conductance (μmol m$^{-2}$ s$^{-1}$); $r_s$ is the leaf stomatal resistance (s m$^2$ μmol$^{-1}$); $m$ is an empirical parameter that relates stomatal conductance and photosynthesis with values ranging from 5 to 9; $A$ is the photosynthesis rate as described above; $c_s$ is the $CO_2$ partial pressure at the leaf surface (Pa); $e_s$ is the vapor pressure at the leaf surface (Pa); $e_i$ is the saturation vapor pressure inside the leaf (Pa); $P_{atm}$ is the atmospheric pressure (Pa); and $b$ is the minimum stomatal conductance.

As mentioned above, following Lombardozzi et al. (2015), an $O_3$ damage scheme was implemented in Noah-MP embedded in WRF-Chem model version 3.8.1. The photosynthesis rate and stomatal conductance are modified independently using two sets of $O_3$ impact factors, $F_{pO_3}$ and $F_{cO_3}$, respectively, which are then multiplied to the initial $A$ and $g_s$ calculated by the Farquhar-Ball-Berry model, respectively. Lombardozzi et al. (2012) found that independently modifying stomatal conductance and photosynthesis can improve the model prediction of plant response to $O_3$ damage. The two damage factors are calculated based on the cumulative uptake of $O_3$ (CUO), which integrates the $O_3$ flux inside leaves through the stomata throughout the growing season. The CUO (mmol m$^{-2}$) is calculated as

$$\text{CUO} = 10^{-6}\sum\frac{[O_3]}{k_{O_3}r_s + r_a + r_b}\Delta t \tag{3}$$

Where $[O_3]$ is the surface $O_3$ concentration (nmol m$^{-3}$); $k_{O_3} = 1.61$ is the ratio of leaf resistance to $O_3$ to leaf resistance to water (Uddling et al., 2012); $r_s$ is the stomatal resistance, $r_a$ is the aerodynamic resistance and $r_b$ is the boundary-layer resistance (s m$^{-1}$); $\Delta t$ is the model time step (s). CUO is only accumulated when LAI is larger than 0.4 and $O_3$ flux is larger than a threshold value of 0.8 nmol $O_3$ m$^{-2}$ s$^{-1}$ to consider the detoxification effect of plants to $O_3$ damage.

The two damage factors have linear relationships with CUO and can be calculated as follows:

$$F_{pO_3} = a_p \times \text{CUO} + b_p \tag{4}$$
$$F_{cO_3} = a_c \times \text{CUO} + b_c \tag{5}$$

where $F_{pO_3}$ is the $O_3$ damage factor for photosynthesis and $F_{cO_3}$ is the $O_3$ damage factor for stomatal conductance; $a_p$, $b_p$, $a_c$, and $b_c$ are empirical slopes and intercepts of three different plant groups (broadleaf trees, needleleaf trees, and grasses or crops) from Lombardozzi et al. (2015). The values of

these slopes and intercepts are shown in Table 1. The original photosynthesis and stomatal conductance are then multiplied with the two damage factors, respectively to get the modified photosynthesis and stomatal conductance under $O_3$ exposure.

**2.3 Model Experiments and Evaluation**

Two sets of experiments were conducted in this study. We performed a control simulation (simu_without$O_3$) without $O_3$ damage on vegetation and a production simulation (simu_with$O_3$) with $O_3$ damage on vegetation. Detailed information of the experiments is shown in Table 2. In the simu_with$O_3$ experiment, the $O_3$ concentration simulated by the chemical module of the model is dynamically passed onto the land surface model at every time step to modify the photosynthesis and stomatal conductance. The differences between the two sets of experiments including vegetation physiology, meteorological fields and $O_3$ concentration can thus be attributed to $O_3$-vegetation interactions. In this work, each simulation was conducted from 24 May to 1 September every year from 2014 to 2017 and the days in May were discarded as spin-up. For each simulation in the four years, anthropogenic emissions were kept at 2014 levels, while meteorological fields were changing every year. The 4-year June-July-August (JJA) averaged results were analyzed and compared. JJA was selected because of the most severe $O_3$ pollution in this season and because it is within the active growing season of the plants.

The simulated meteorological variables and air pollutant concentrations were evaluated using available in-situ observations in China. The daily meteorological observations including temperature at 2 meter ($T_{2m}$), relative humidity at 2 meter ($RH_{2m}$), and wind speed at 10 meter ($WS_{10m}$) above displacement height were from the National Meteorological Information Center. There are 698 stations in the study domain. The air pollutant observations were provided by the China National Environmental Monitoring Center (CNEMC) network, which offers hourly concentrations of particulate matter with an aerodynamic diameter of less than 2.5 μm ($PM_{2.5}$) and 10 μm ($PM_{10}$), carbon monoxide (CO), $O_3$, sulfur dioxide ($SO_2$) and nitrogen dioxide ($NO_2$). The locations of meteorological stations and the sites of CNEMC network are shown in Figure 2. The statistical parameters including mean values (Mean) of observations and simulated variables, their standard deviations (SD), indices of agreement (IOA), mean biases (MB), and correlation coefficients (CORR) were computed to evaluate the model performance in this study.

**3.  Results**

**3.1   Model evaluation**

Table 3 shows the city-averaged evaluation results of meteorological variables from the modified model. The information of the major cities used for evaluation is shown in Table S4. From Table 3, we can find that $T_{2m}$ is underestimated with MB values ranging from −1.00 °C in 2017 to −0.70 °C in 2014. The IOA and CORR are generally higher than 0.8, indicating that the model could reasonably simulate the variations of $T_{2m}$. Unlike temperature, relative humidity is overestimated by the model simulations with MB values ranging from 4.38 in year 2014 to 7.33 in year 2016, but the CORR values with observations are still high (CORR > 0.7). Wind speed is also overestimated by more than 0.38 m s$^{-1}$, which might be caused by the underestimation of terrain height as reported in other WRF modeling studies (Brunner et al., 2015; Liu et al., 2020). The detailed evaluation results for each city and for seven major geographic

regions of China are shown in Table S5-S10. The classification of the geographic regions is shown in Fig. S2. As shown in these tables, the model can reasonably capture the spatial distribution of these meteorological variables. For example, the larger values of $T_{2m}$ and $RH_{2m}$ in cities from southern China compared with the cities in northern China (Table 4) can be reasonably simulated. We also found that the model simulations have better performance in northeastern China, central China and southern China in terms of IOA and CORR as shown in these tables (Table 4).

Table 5 shows the city-averaged evaluation results of six air pollutants simulated from the modified model. The information of the major cities used for air pollutant evaluation is shown in Table S11. Form Table 5, positive MB values for $O_3$, $PM_{2.5}$, $SO_2$, and $NO_2$, and negative MB values for CO are found. The overestimation of $O_3$ by WRF-Chem was also reported by Hu et al. (2016) and Gao et al (2020). For $PM_{10}$, both positive and negative MB values are found for different years. The results indicate general overestimation by the model of most air pollutants except for CO. The underestimation of CO can be explained by either $O_3$ chemistry, which points to the problem related to low titration, or in the underestimation of dry deposition by the model, which is also affected by the modification of the model. The IOA of air pollutant concentration ranges from 0.36 ($SO_2$) to 0.63 ($O_3$). The correlation coefficient of air pollutants ranges from 0.14 ($PM_{10}$) to 0.66 ($O_3$). Detailed evaluation results for each city and major geographic regions of China are shown in Tables S9–S14 and Table 6. In terms of the evaluation for $O_3$, the model has better performance in northeastern China, eastern and southern China, which may suffer the most severe $O_3$ damage. Our results are generally consistent with the evaluation results of CMAQ simulation over China by Liu et al. (2020). MBs of $SO_2$, $NO_2$ and CO are consistent in both magnitude and sign with Liu et al. (2020), while the MBs of PM and $O_3$ are larger than Liu et al. (2020). Correlation coefficients of air pollutants are also of similar magnitude with Liu et al. (2020), showing that our model results can well capture the temporal variations of air pollutants. We also compared the evaluation results between the original model and the modified model, as shown in Table S2 and Table S3 in the supplement and Table 3 and Table 5 here. We found no obvious differences in the evaluation results between the original model results and the revised model results. It should be noted that this study might not be able to and was not meant to improve model accuracy, but our modified model is able to capture $O_3$-vegetation interactions without worsening model performance. Overall, there are systematic biases in simulated variables especially the air pollutant concentrations, but the spatial distribution of both meteorological variables and air pollutant concentrations are reasonably simulated by the model, lending trust to the use of the model for sensitivity studies to examine the effects of $O_3$-vegetation interactions on the atmospheric environment.

### 3.2    Responses of vegetation to $O_3$ damage

$O_3$ can adversely affect photosynthesis rate and stomatal conductance and therefore interfere with vegetation growth, productivity and transpiration. To understand the $O_3$-induced damage on vegetation physiology, the spatial distribution and changes in stomatal resistance (RS), photosynthesis rate (PSN), LAI, GPP, and transpiration rate (TR) during 2014–2017 summer (June-July-August) were analyzed.

Figure 3a and 3d display the spatial distribution of sunlit stomatal resistance (RSSUN) and shaded stomatal resistance (RSSHA) from the simu_withoutO$_3$ experiment, respectively. The absolute and relative changes in RSSUN (RSSHA) between simu_withO$_3$ and simu_withoutO$_3$ experiments are shown

in the middle and the right panel of Fig. 3, separately. In general, simulated stomatal resistance in eastern China is larger than that in western China. Both RSSUN and the RSSHA are enhanced in response to $O_3$ damage to vegetation. The maximum increases in RSSUN and RSSHA can be up to $1.0 \times 10^3$ s m$^{-1}$, which is equivalent to a ~16% increase compared to the simu_withoutO$_3$ simulation. Comparing the changes in RSSUN vs. RSSHA, the changes in RSSHA are larger than that in RSSUN, reflecting the larger sensitivity of shaded leaves to $O_3$ damage (Kinose et al., 2017). Northern China experiences larger changes in stomatal resistance generally, especially in Henan, Hebei, and Shandong provinces, where the changes in stomatal resistance are twice as much as the changes in stomatal resistance over other regions.

The spatial distribution of 2014–2017 JJA mean PSN, LAI and GPP from the simu_withoutO$_3$ simulations and their changes induced by $O_3$ damage are presented in Fig. 4. From Fig. 4a, we find that the PSN values are generally higher in eastern China compared with western China with the largest values of up to ~7 $\mu$mol CO$_2^{-1}$ m$^{-2}$ s$^{-1}$. Similar spatial distribution and hotspot areas can also be observed for LAI (Fig. 4d) and GPP (Fig. 4g), with LAI and GPP values in hotspot areas up to 3.6 and 10 g C m$^{-2}$ day$^{-1}$, respectively. We also find that Henan, Hebei, Shanxi and Shandong provinces have smaller values in PSN, LAI and GPP when compared with other provinces in eastern China.

With $O_3$ damage, PSN decreases in general, with absolute changes in PSN ranging from 0.6 to 3.6 $\mu$mol CO$_2$ m$^{-2}$ s$^{-1}$ (Fig. 4b), representing 20–40% reductions in PSN. For northeastern and southern China, where the original PSN values are large, ~20% reductions in PSN are found (Fig. 4c). In western China where the dominant vegetation type is grassland and the original PSN values are small, more than 40% of PSN is reduced due to $O_3$ damage (Fig. 4c). In response to the PSN reductions, LAI and GPP also decrease. More than 0.4 reductions in LAI are found in central and northern China (Fig. 4e), corresponding to more than 20% reductions in LAI; in other regions, 5–15% reductions in LAI are observed. More than 0.8 g C m$^{-2}$ day$^{-1}$ reductions in GPP are found generally in China. Similar to Fig. 3c, we find that GPP decreases by ~20% in northeastern and southern China and decreases by more than 40% in other regions (Fig. 4i). Based on offline models without considering atmosphere-biosphere coupling, $O_3$ damage was found to decrease GPP at most by 11–17% in the East Coast hotspots of the US (Yue and Unger, 2014). Using the offline CLM model, Lombardozzi et al. (2015) estimated that the present $O_3$ exposure reduces GPP globally by 8–12%. Based on RegCM-CHEM4 regional climate model coupled with YIBs terrestrial biosphere model, Xie et al. (2019) revealed that $O_3$ damage induces a significant reduction (12.1±4.4%) in the GPP, up to 35% in summer over China (Table S15). Comparing our results with previous studies, our results are broadly consistent with Xie et al. (2019) but the magnitude is larger than the studies conducted by Yue and Unger (2014) and Lombardozzi et al. (2015). Differences or uncertainties may arise from the different model settings. It appears that offline models as used by Yue and Unger (2014) and Lombardozzi et al. (2015) generally found smaller damage than studies with two-way coupling between the atmosphere and biosphere as used by Xie et al. (2019) and our work; this could be due to the existence of positive biosphere-atmosphere feedbacks that potentially worsen $O_3$ damage, as will be discussed in subsequent sections. Different $O_3$ damage schemes employed in the models may also be a source of differences, although we note that both this work and Lombardozzi et al. (2015) used the same scheme, so the differences appear to arise more likely from the effect of coupling and other model settings than from the schemes alone.

The spatial distribution of dominant vegetation types in China are shown in Fig. 1, where we can see that the croplands dominant in eastern China and especially in southern China suffer the greatest GPP reductions, indicating that crop yields in China would also be heavily affected by $O_3$ damage.

Figure 5 depicts the spatial distribution of transpiration rate (TR) of vegetation and the changes in transpiration rate induced by $O_3$ damage. TR values are higher in eastern China where there is larger vegetation coverage (Fig. 5a). As shown in Fig. 5b, TR deceases by 0.2–1.0 mm day$^{-1}$ generally in eastern China with large reductions in northern China, especially in Henan, Shandong, Anhui and Jiangsu provinces. In terms of relative changes, TR decreases by ~12% in northeastern and southern China, while more than 24% reductions are found in other regions. Transpiration is affected by the changes in both RS and LAI. With $O_3$ damage, both the increases in RS (Fig. 3c and Fig. 3f) and decreases in LAI (Fig. 4f) cause TR to decrease, as shown in Fig. 5b and 5c. Comparing the changes in RS (Fig. 3c and Fig. 3f), LAI (Fig. 4f) and TR (Fig. 5c), we can find that the distribution of changes in TR is more consistent with that of RS, reflecting the dominance of RS in controlling TR.

### 3.3   Changes in meteorology due to $O_3$-vegetation coupling

Through interacting with vegetation, $O_3$ has the potential to further affect the meteorological environment in China via modifying, e.g., surface heat fluxes, temperature, humidity, and boundary layer height. The distribution of meteorological variables from simulations with and without $O_3$ damage is thus compared and analyzed in this section.

Figure 6 shows the spatial distribution of latent heat (LH) flux and sensible heat (SH) flux, and the changes in LH and SH due to $O_3$-vegetation coupling. With $O_3$ included in the model simulations, the LH flux decreases by more than 4 W m$^{-2}$ (Fig. 6b) on average following the decreases in transpiration rate. Hotspot areas are found in Henan, Shandong, Anhui and Jiangsu provinces, where reductions in LH can be up to 30 W m$^{-2}$. Meanwhile, 5–30 W m$^{-2}$ increases in SH flux are observed in central and northern China (Fig. 6d). With $O_3$-vegetation coupling, more than 20% reductions in LH flux are found in central and northern China (Fig. 6c), 20% increment in SH flux are found in similar regions (Fig. 6f), indicating that $O_3$ damage shifts the energy balance toward more net radiation being dissipated by SH flux than LH flux, with ramifications for surface temperature.

Figure 7 shows the distribution and the changes in surface relative humidity, temperature and planetary boundary layer height (PBLH) in response to $O_3$ damage. Reductions in transpiration rate can directly cause reductions in relative humidity. As shown in Fig. 7b, relative humidity has at least 3% absolute reductions. Values of relative humidity decrease more in northern China than in southern China. Similar to the changes in TR (Fig. 5b), larger reductions in relative humidity (3–9%) are found over Henan, Hebei, Shandong, Anhui provinces. The decreases in LH flux and increases in SH flux following the changes in transpiration rate drive the increases in temperature and contribute to PBLH growth. As presented in Fig. 7e and Fig. 7h, the distribution and hotspot areas of the changes in temperature and PBLH are similar to those in relative humidity. Generally, northern China has larger increases of temperature and PBLH compared with other regions. Generally, temperature increases by 0.2–0.8 K and PBLH increases by 40–120 m for northern China. The hotspot areas experience at least 0.6 K increases in temperature, and 80 m increases in PBLH.

As shown in Table S15, our results are comparable with results from a regional simulation conducted by Li et al. (2016), which showed that $O_3$ damage decreases LH flux by 10–27 W m$^{-2}$ and $O_3$ damage increases temperature by 0.6 °C–2.0 °C in the US. However, in their study, Li et al. (2016) assumed that $O_3$ damage to plants happens when $O_3$ concentration is over a threshold of 20 ppb to imitate a weaker detoxifying effect of plants, instead of the 40 ppb threshold that was commonly used in previous studies.

Considering the severe $O_3$ air pollution in China, we resorted to use the more universal $O_3$ threshold used by previous studies (Lombardozzi et al., 2015; Sadiq et al., 2017; Zhou et al., 2018) to represent a more conventional detoxifying effect, instead of lowering the threshold value that would cause much larger changes in the surface fluxes and meteorological fields. Using a two-way coupling model and the same $O_3$ damage scheme, Arnold et al. (2018) revealed that $O_3$ causes less than 8 W m$^{-2}$ changes in surface

heat fluxes regionally, which is smaller than the changes of surface heat fluxes in our study. One possible reason is that the simulated changes in $O_3$ and aerosol in Arnold et al. (2018) did not feedback onto radiation and climate simulation or affect LAI.

### 3.4    $O_3$-vegetation feedbacks on $O_3$ concentrations


$O_3$-induced changes in vegetation, surface fluxes and the overlying meteorology can also constitute important feedback effects onto $O_3$ concentration itself. Figure 8 shows the spatial distribution of surface $O_3$ concentration. The change in surface $O_3$ concentration during daytime is also shown in Fig. S2. As shown in Fig. 8a (Fig. S2), surface $O_3$ concentration is higher in central and northern China during

summer. In terms of the feedbacks on $O_3$ concentration, we found generally enhancements in $O_3$ concentration when $O_3$-vegetation interactions are accounted for, thus representing a positive feedback that worsens $O_3$ air quality (Fig. 8b). $O_3$ concentration increases the most (by up to 6 %) in Hebei, Shanxi and Henan provinces, with the maximum increment of 6 ppb. The enhancement in surface $O_3$ concentration from our study is at the similar magnitude with that from the study conducted by Sadiq et

al. (2017), in which both biogeochemical and meteorological feedbacks from $O_3$-vegetation interactions to $O_3$ are considered. Without considering the meteorological feedbacks following the changes in transpiration to $O_3$ concentrations, smaller feedbacks on surface $O_3$ concentrations are found by the following studies. For instance, by incorporating $O_3$-LAI coupling in chemical transport model, Zhou et al. (2018) found an $O_3$ feedback of −1.8 to +3 ppb globally. Another similar work conducted by Gong et

al. (2020) showed that $O_3$-induced inhibition in stomatal conductance increases surface $O_3$ by 2.1 ppb in eastern China, while considering the addition effects of $O_3$ on isoprene emission slightly reduces surface $O_3$ concentrations by influencing the precursors. Soil moisture deficit, which has been shown to reduce stomatal uptake, if considered, will also contribute to the enhancement in $O_3$ concentration (Rydsaa et al., 2016). Together with previous findings, it is increasingly clear that meteorological feedback could

be an important pathway whereby $O_3$-vegetation interactions can further worsen $O_3$ air quality, almost doubling the effect of biogeochemical feedback alone (i.e., via changes in $O_3$-relevant chemical fluxes alone). It should be cautiously noted that in terms of magnitude alone the model biases in $O_3$ are comparable and sometimes larger than the up to 6 ppb systematic enhancement caused by $O_3$ damage, which represents be one major source of uncertainties in our study.


Reduced dry deposition due to stomatal closure and reduced LAI, as well as increased isoprene emission, are all found to be the drivers for the overall positive $O_3$ feedback. Reductions in dry deposition velocity,

following closely the corresponding reductions in transpiration rate as both processes are modulated by stomatal regulation, contribute in part to the $O_3$ enhancement. Figure 9 shows the spatial distribution of isoprene emission and its changes due to $O_3$ damage. We observe general increases in isoprene emission in eastern China, mainly due to increased surface temperature (Figs. 7e and 7f) that is more than enough to offset reduced isoprene caused by reduced LAI (Figs 4e and 4f). All in all, $O_3$ damage on vegetation can further enhance $O_3$ levels via an overall positive effect, due to not only the associated reductions in dry deposition velocity, but also the reductions in transpiration, LH flux and the resulting rise in surface temperature.

**4    Conclusions**

Tropospheric $O_3$ is one of the most concerning air pollutants due to its global warming effects and its ability to affect human health, vegetation and crops. $O_3$ and vegetation closely interact with each other and such interactions may not only affect plant physiology (e.g., stomatal conductance and photosynthesis) but also influence the overlying meteorology and air quality through modifying leaf stomatal behavior, plant structure (e.g., LAI) and subsequently land-atmosphere fluxes. According to previous field experiments and modeling works, China has been recognized as one of the hotspot areas suffering from severe $O_3$ pollution and the resulting damage on vegetation and crops, but the feedback effects onto air quality and climate have not been fully characterized. Previous studies mainly focused on the global scale with coarse spatial resolutions, which did not fully capture the spatial distribution of $O_3$ damage on vegetation in China. Based on the results from global studies pointing out that China is a hotspot in terms of $O_3$ pollution and $O_3$ damage on vegetation, our model simulations performed at high spatial resolutions were capable of investigating $O_3$ damage effects on regional and provincial scales in China. In this study, we examined the effects of $O_3$-vegetation interactions on $O_3$ air quality and meteorology in China during 2014–2017 based on the two-way coupled WRF-Chem model simulations whereby $O_3$, meteorology and vegetation physiology and structure can co-evolve with each other in real time.

We found that in China stomatal resistance is enhanced by up to 16%, which is the direct response to $O_3$ damage. Northern China, especially Henan, Hebei, and Shandong provinces, is identified as a hotspot area. For photosynthesis, more than 20% reductions are observed in China. Large reductions (>2.4 μmol $CO_2$ m$^{-2}$ s$^{-1}$) are found in northeastern and southern China. Following reduced photosynthesis, LAI shows relatively small reductions (5–15%), while GPP shows more than 20% reductions (1.6 g C m$^{-2}$ day$^{-1}$). Changes in transpiration rate are due to both changes in stomatal resistance and changes in LAI. With the increases in stomatal resistance and decreases in LAI, transpiration deceases from 0.2 to 1.0 mm day$^{-1}$ in eastern China with the largest reductions occur in northern China. We also found that the distribution of changes in transpiration is consistent more with the distribution of stomatal resistance than with those of LAI, indicating the dominance of the former in contributing to the overall transpiration rate.

With $O_3$ damage, the LH fluxes decrease by more than 4 W m$^{-2}$ on average, with hotspot areas appearing in Shandong, Anhui and Jiangsu provinces, in which the decreases can be up to 30 W m$^{-2}$ following mostly the decreases in transpiration rate. SH fluxes increase in similar areas at comparable magnitudes (10–25 W m$^{-2}$). The decreases in LH and the increases in SH cause the increases in temperature and PBLH. We found that northern China has larger decreases in relative humidity, temperature and PBLH

compared with other regions. Generally, relative humidity shows at least 4% relative reductions, temperature increases by 0.2–0.8 K, and PBLH increases by 40–120 m for northern China. This indicates that $O_3$-vegetation interactions will cause a shift in the energy balance toward a state where available net radiation is dissipated more by SH flux than LH flux, with ramifications for surface temperature. This represents an additional pathway whereby anthropogenic $O_3$ pollution can worsen warming, in addition to $O_3$ being a greenhouse gas itself and $O_3$-induced plant damage diminishing the global net carbon sink (e.g., Sitch et al., 2007; Lombardozzi et al., 2015).

$O_3$ induces changes in vegetation, surface fluxes and meteorology, and in turn affects its own concentration. In this study, we found that reduced dry deposition in China is mainly due to enhanced stomatal conductance, while enhanced isoprene emission is mainly due to enhanced surface temperature and the corresponding increase in $O_3$ concentration. $O_3$ concentration increases the most (up to 6%) in Hebei, Shanxi and Henan provinces, with the maximum value of 6 ppb. Our results demonstrate that $O_3$-vegetation interactions can lead to strong positive feedback that can amplify $O_3$ pollution in China, in agreement with the suggestions by previous studies focusing on a global scale (Sadiq et al., 2017; Zhou et al., 2018; Gong et al., 2020). We also found that fully considering the positive $O_3$-vegetation feedbacks, especially when meteorological changes are also accounted for, generates greater damage on vegetation productivity than found by studies that only considered "offline" $O_3$ damage on plants without feedbacks (Yue and Unger, 2014; Lombardozzi et al., 2015).

In this study, the summertime simulation period of JJA was selected due to the high $O_3$ pollution in this season and the overlapping with vegetation growing season to capture the severe $O_3$ damage on vegetation. Nevertheless, uncertainty may still arise from that our simulation period may not cover the growing season of all vegetation types and may not cover all periods that $O_3$ damage happens, which may represent an underestimation of the full scale of $O_3$ damage. Future work should be conducted for longer time periods and for all seasons, which will help us better understand $O_3$-vegetation interactions in China. Uncertainty may also arise from the $O_3$ scheme employed in this study in terms of the CUO calculation and the consideration of $O_3$ detoxification mechanism of different vegetation types. The calculation of CUO heavily relies on the $O_3$ threshold. Considering the sensitivies of different vegetation types to $O_3$ damage, CUO threshold should be varied with different vegetation types. However, a constant $O_3$ threshold was employed in our study for the whole simulation domain and for all vegetation types, which may either underestimate or overestimate the actual $O_3$ damage. Moreover, following the work of Lombardozzi et al. (2015), we classified all the vegetation types into only three groups, which may be too coarse to investigate $O_3$ damage effects on regional or local scales. For example, Zhou et al. (2018) pointed out that Lombardozzi et al. (2015) treated tropical and temperate plants equivalently, which might lead to possible biases. More studies should be conducted to derive more appropriate $O_3$ thresholds for CUO calculation and make them available for regional scales or for different vegetation types. Another source of uncertainty may arise from the lack of representation of the direct effect of $O_3$ on isoprene emission. As pointed out by Gong et al. (2020), including the effect of $O_3$ damage on isoprene emission may reduce $O_3$ concentration by influencing precursors, but increase $O_3$ concentration at the same time through weakening the shortwave radiative forcing of secondary organic aerosols, which would help constitute a more complete feedback mechanism between $O_3$ and vegetation. Moreover, uncertainties may also come from that the effect of soil moisture deficit was not considered in this study, which may underestimate the reduction in dry deposition sink of $O_3$. It should also be noted that keeping

the anthropogenic emission inventory fixed in 2014 levels may be another limitation because of the nonlinear chemistry involving biogenic and anthropogenic precursors. Despite these uncertainties and limitations, our study provides detailed and comprehensive results whereby $O_3$-vegetation impacts will adversely affect plant growth and crop production, contribute to global warming, worsen the severe $O_3$

air pollution in China via feedbacks, and identifies the hotspot areas in the country. Our findings clearly pinpoint the need to consider the $O_3$ damage effects in both air quality studies and climate change studies.

*Data availability.* Model output data used for analysis and plotting, and the code used for simulations can be made available upon request.


*Author contributions.* APKT and SHLY conceived the study. JCZ carried out the model simulations and drafted the manuscript. SHLY and APKT supervised and funded the study.

*Competing interests.* The authors declare that they have no conflict of interest.


**Acknowledgments**

This work is jointly funded by the Vice-Chancellor's Discretionary Fund of The Chinese University of Hong Kong (grant no. 4930744) given to both Steve H.L. Yim and Amos P. K. Tai, by Dr. Stanley Ho

Medicine Development Foundation (grant no. 8305509) given to Steve H.L. Yim, and by a Research Grants Council (Hong Kong) General Research Fund grant (grant no. 14306220) given to Amos P. K. Tai. We would like to thank Dr. Jialun Li for the sharing and guidance of WRF-Chem code implementation in this work.

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

**Table 1.** Slopes (per mmol m$^{-2}$) and intercepts (unitless) used for O$_3$ damage factors in Eqs. (4) and (5), following Lombardozzi et al. (2015).


| | Photosynthesis | | Conductance | |
|---|---|---|---|---|
| | Slope ($a_p$) | Intercept ($b_p$) | Slope ($a_c$) | Intercept ($b_c$) |
| Broadleaf | 0.0000 | 0.8752 | 0.0000 | 0.9125 |
| Needleleaf | 0.0000 | 0.8390 | 0.0048 | 0.7823 |
| Grasses and crops | −0.0009 | 0.8021 | 0.0000 | 0.7511 |

**Table 2.** Description of the two sets of model experiments.

| Experiment name | Year | Anthropogenic Emission | Meteorological ICs and BCs |
|---|---|---|---|
| simu_withoutO$_3$ | 2014–2017 JJA | Year 2014 | FNL |
| simu_withO$_3$ | 2014–2017 JJA | Year 2014 | FNL |


**Table 3.** Evaluation results for the temperature at 2 meter ($T_{2m}$), relative humidity at 2 meter (RH$_{2m}$) and wind speed at 10 meter (WS$_{10m}$) for different years in China. Mean_obs (Mean_simu) is the mean value of observation (model simulation); SD_obs (SD_simu) is the standard deviation of the observation (model simulation); IOA is the index of agreement; CORR is the correlation coefficient; MB is the mean bias.

| | Year | Mean_obs | SD_obs | Mean_simu | SD_simu | IOA | CORR | MB |
|---|---|---|---|---|---|---|---|---|
| $T_{2m}$ | 2014 | 25.41 | 2.61 | 24.71 | 2.27 | 0.86 | 0.87 | −0.70 |
| (℃) | 2015 | 25.41 | 2.56 | 24.67 | 2.24 | 0.86 | 0.89 | −0.74 |
| | 2016 | 26.35 | 2.82 | 25.44 | 2.61 | 0.85 | 0.85 | −0.91 |
| | 2017 | 26.29 | 3.17 | 25.28 | 3.16 | 0.81 | 0.78 | −1.00 |
| RH$_{2m}$ | 2014 | 74.77 | 10.22 | 79.14 | 8.96 | 0.67 | 0.71 | 4.38 |
| (%) | 2015 | 73.34 | 10.75 | 80.50 | 8.73 | 0.68 | 0.75 | 7.16 |
| | 2016 | 74.14 | 10.81 | 81.47 | 10.10 | 0.70 | 0.73 | 7.33 |
| | 2017 | 73.24 | 11.65 | 79.89 | 9.62 | 0.68 | 0.69 | 6.63 |
| WS$_{10m}$ | 2014 | 1.84 | 0.66 | 2.22 | 1.16 | 0.54 | 0.40 | 0.38 |
| (m s$^{-1}$) | 2015 | 2.00 | 0.74 | 2.48 | 1.35 | 0.55 | 0.44 | 0.48 |
| | 2016 | 1.99 | 0.70 | 2.47 | 1.32 | 0.54 | 0.45 | 0.48 |
| | 2017 | 2.02 | 0.72 | 2.51 | 1.42 | 0.53 | 0.45 | 0.50 |

**Table 4.** Evaluation results of temperature at 2 meter ($T_{2m}$), relative humidity at 2 meter ($RH_{2m}$) and wind speed at 10 meter ($WS_{10m}$) in 7 major geographic regions from the implemented model. NE is northeast China, NC is north China, CC is central China, EC is east China, SC is south China, SW indicates southwest China, and NW is northwest China. Mean_obs (Mean_simu) is the mean value of observations (model simulations); SD_obs (SD_simu) is the standard deviation of the observations (model simulations); IOA is the index of agreement; CORR is the correlation coefficient; MB is the mean bias.

|  | Region | Mean_obs | SD_obs | Mean_simu | SD_simu | IOA | CORR | MB |
|---|---|---|---|---|---|---|---|---|
| $T_{2m}$ | NEC | 23.01 | 3.05 | 22.73 | 3.01 | 0.94 | 0.91 | -0.28 |
| (℃) | NC | 24.94 | 2.76 | 25.84 | 2.84 | 0.86 | 0.88 | 0.88 |
|  | CC | 27.62 | 3.05 | 26.87 | 2.75 | 0.92 | 0.88 | -0.75 |
|  | EC | 27.33 | 2.99 | 26.46 | 2.58 | 0.90 | 0.89 | -0.87 |
|  | SC | 28.60 | 1.49 | 28.61 | 1.30 | 0.75 | 0.64 | 0.01 |
|  | SWC | 23.20 | 2.32 | 21.61 | 2.14 | 0.77 | 0.80 | -1.58 |
|  | NWC | 20.20 | 2.87 | 18.55 | 3.01 | 0.77 | 0.89 | -1.65 |
| $RH_{2m}$ | NEC | 71.7 | 11.49 | 71.98 | 14.00 | 0.85 | 0.79 | 0.93 |
| (%) | NC | 63.25 | 13.94 | 57.01 | 14.08 | 0.79 | 0.75 | -6.24 |
|  | CC | 79.23 | 10.11 | 88.29 | 8.61 | 0.70 | 0.71 | 9.06 |
|  | EC | 78.93 | 9.99 | 88.80 | 8.31 | 0.69 | 0.79 | 9.87 |
|  | SC | 81.26 | 6.54 | 88.41 | 5.66 | 0.62 | 0.60 | 7.14 |
|  | SWC | 78.92 | 9.11 | 93.34 | 5.13 | 0.52 | 0.64 | 13.40 |
|  | NWC | 57.93 | 13.34 | 58.48 | 14.10 | 0.75 | 0.76 | 0.55 |
| $WS_{10m}$ | NEC | 2.22 | 0.93 | 3.08 | 1.80 | 0.62 | 0.62 | 0.86 |
| (m s$^{-1}$) | NC | 2.06 | 0.72 | 2.45 | 1.29 | 0.57 | 0.48 | 0.38 |
|  | CC | 2.06 | 0.81 | 2.38 | 1.41 | 0.61 | 0.51 | 0.33 |
|  | EC | 2.18 | 0.76 | 2.85 | 1.54 | 0.59 | 0.55 | 0.67 |
|  | SC | 2.02 | 0.76 | 2.81 | 1.51 | 0.52 | 0.43 | 0.80 |
|  | SWC | 2.16 | 0.76 | 2.54 | 1.40 | 0.57 | 0.51 | 0.37 |
|  | NWC | 1.46 | 0.50 | 2.91 | 1.38 | 0.30 | 0.23 | 1.45 |





**Table 5.** Evaluation results for the air pollutants in China. Mean_obs (Mean_simu) is the mean value of observation (model simulation); SD_obs (SD_simu) is the standard deviation of the observation (model simulation); IOA is the index of agreement; CORR is the correlation coefficient; MB is the mean bias.

| | Year | Mean_obs | SD_obs | Mean_simu | SD_simu | IOA | CORR | MB |
|---|---|---|---|---|---|---|---|---|
| $O_3$ | 2014 | 29.79 | 9.95 | 51.49 | 18.60 | 0.48 | 0.57 | 22.13 |
| (ppb) | 2015 | 32.04 | 10.16 | 48.98 | 18.27 | 0.54 | 0.55 | 16.95 |
| | 2016 | 33.28 | 10.59 | 48.47 | 18.18 | 0.56 | 0.58 | 15.14 |
| | 2017 | 35.74 | 11.71 | 49.50 | 19.61 | 0.63 | 0.66 | 13.82 |
| $PM_{2.5}$ | 2014 | 46.30 | 21.52 | 63.28 | 27.15 | 0.52 | 0.33 | 18.61 |
| ($\mu g\ m^{-3}$) | 2015 | 38.52 | 17.30 | 55.56 | 24.85 | 0.55 | 0.42 | 16.66 |
| | 2016 | 31.86 | 13.96 | 56.70 | 25.69 | 0.47 | 0.40 | 24.54 |
| | 2017 | 28.82 | 12.23 | 56.34 | 25.70 | 0.40 | 0.30 | 27.65 |
| $PM_{10}$ | 2014 | 80.79 | 31.62 | 71.74 | 28.65 | 0.47 | 0.22 | −7.51 |
| ($\mu g\ m^{-3}$) | 2015 | 72.03 | 29.74 | 63.83 | 26.29 | 0.50 | 0.26 | −8.93 |
| | 2016 | 59.68 | 22.21 | 65.01 | 27.29 | 0.49 | 0.24 | 4.65 |
| | 2017 | 57.83 | 22.18 | 64.78 | 27.25 | 0.41 | 0.14 | 6.95 |
| $SO_2$ | 2014 | 6.11 | 2.36 | 8.41 | 3.22 | 0.48 | 0.41 | 2.36 |
| (ppb) | 2015 | 4.78 | 1.89 | 8.39 | 3.26 | 0.44 | 0.45 | 3.64 |
| | 2016 | 4.17 | 1.57 | 8.08 | 3.16 | 0.41 | 0.36 | 3.92 |
| | 2017 | 3.83 | 1.33 | 8.58 | 3.52 | 0.36 | 0.42 | 4.78 |
| $NO_2$ | 2014 | 17.20 | 4.51 | 17.23 | 4.63 | 0.41 | 0.26 | 0.06 |
| (ppb) | 2015 | 16.01 | 4.47 | 17.37 | 4.98 | 0.43 | 0.31 | 1.43 |
| | 2016 | 15.29 | 4.29 | 17.35 | 5.11 | 0.43 | 0.31 | 2.06 |
| | 2017 | 15.83 | 4.37 | 17.84 | 5.12 | 0.43 | 0.32 | 2.02 |
| CO | 2014 | 0.76 | 0.19 | 0.44 | 0.11 | 0.48 | 0.42 | −0.32 |
| (ppm) | 2015 | 0.67 | 0.15 | 0.45 | 0.11 | 0.49 | 0.42 | −0.22 |
| | 2016 | 0.65 | 0.14 | 0.45 | 0.11 | 0.50 | 0.45 | −0.20 |
| | 2017 | 0.64 | 0.12 | 0.46 | 0.11 | 0.47 | 0.38 | −0.18 |



**Table 6.** Evaluation results of air pollutants in 7 major geographic regions simulated by the implemented model. NE is northeast China, NC is north China, CC is central China, EC is east China, SC is south China, SW indicates southwest China, and NW is northwest China. Mean_obs (Mean_simu) is the mean value of observations (model simulations); SD_obs (SD_simu) is the standard deviation of the observations (model simulations); IOA is the index of agreement; CORR is the correlation coefficient; 845 MB is the mean bias.

|  | Region | Mean_obs | SD_obs | Mean_simu | SD_simu | IOA | CORR | MB |
|---|---|---|---|---|---|---|---|---|
| **O$_3$** | NEC | 32.49 | 10.54 | 44.54 | 15.70 | 0.64 | 0.64 | 11.47 |
| **(ppb)** | NC | 38.56 | 10.81 | 70.59 | 25.11 | 0.40 | 0.55 | 32.02 |
|  | CC | 31.68 | 11.20 | 57.13 | 18.17 | 0.47 | 0.60 | 26.82 |
|  | EC | 29.67 | 10.82 | 40.53 | 18.99 | 0.60 | 0.61 | 11.21 |
|  | SC | 19.90 | 8.40 | 34.21 | 14.68 | 0.53 | 0.64 | 14.90 |
|  | SWC | 24.27 | 9.12 | 42.07 | 13.19 | 0.47 | 0.50 | 18.80 |
|  | NWC | 26.63 | 8.12 | 51.65 | 13.70 | 0.34 | 0.42 | 25.58 |
| **PM$_{2.5}$** | NEC | 42.66 | 25.15 | 43.33 | 19.28 | 0.57 | 0.39 | −0.83 |
| **(µg m$^{-3}$)** | NC | 61.60 | 28.28 | 66.83 | 27.81 | 0.68 | 0.52 | 7.03 |
|  | CC | 52.11 | 30.55 | 94.27 | 39.40 | 0.35 | 0.11 | 45.50 |
|  | EC | 52.87 | 25.71 | 87.37 | 38.97 | 0.50 | 0.39 | 36.63 |
|  | SC | 22.58 | 10.16 | 28.62 | 15.17 | 0.67 | 0.57 | 6.92 |
|  | SWC | 32.82 | 12.22 | 76.69 | 33.49 | 0.27 | 0.08 | 47.55 |
|  | NWC | 45.27 | 14.54 | 42.80 | 14.07 | 0.39 | 0.03 | −1.45 |
| **PM$_{10}$** | NEC | 79.68 | 36.48 | 48.99 | 20.64 | 0.49 | 0.32 | −32.25 |
| **(µg m$^{-3}$)** | NC | 111.17 | 39.69 | 74.29 | 29.33 | 0.54 | 0.38 | −35.90 |
|  | CC | 84.80 | 41.01 | 107.65 | 41.51 | 0.37 | 0.05 | 26.30 |
|  | EC | 78.16 | 35.64 | 99.51 | 40.90 | 0.54 | 0.34 | 23.64 |
|  | SC | 43.64 | 15.72 | 34.11 | 16.14 | 0.58 | 0.47 | −8.54 |
|  | SWC | 58.84 | 20.15 | 87.07 | 35.49 | 0.31 | −0.07 | 32.17 |
|  | NWC | 88.54 | 28.17 | 47.77 | 14.72 | 0.35 | −0.13 | −39.68 |
| **SO$_2$** | NEC | 4.91 | 1.95 | 5.10 | 2.55 | 0.60 | 0.42 | 0.27 |
| **(ppb)** | NC | 8.69 | 3.52 | 8.12 | 3.20 | 0.54 | 0.40 | −0.57 |
|  | CC | 7.34 | 2.09 | 14.56 | 5.45 | 0.36 | 0.47 | 7.23 |
|  | EC | 5.39 | 2.24 | 7.86 | 3.19 | 0.57 | 0.53 | 2.40 |
|  | SC | 3.50 | 0.89 | 4.15 | 1.52 | 0.42 | 0.50 | 0.71 |
|  | SWC | 4.74 | 1.93 | 15.71 | 5.12 | 0.31 | 0.13 | 11.42 |
|  | NWC | 6.65 | 2.90 | 4.31 | 1.50 | 0.46 | 0.34 | −2.28 |
| **NO$_2$** | NEC | 19.51 | 4.84 | 14.07 | 5.27 | 0.41 | 0.11 | −5.66 |
| **(ppb)** | NC | 19.57 | 5.13 | 14.05 | 4.14 | 0.48 | 0.27 | −5.61 |
|  | CC | 16.75 | 4.32 | 19.70 | 5.57 | 0.38 | 0.32 | 2.83 |
|  | EC | 16.24 | 4.78 | 28.83 | 6.88 | 0.40 | 0.39 | 12.65 |
|  | SC | 13.23 | 3.48 | 14.02 | 3.96 | 0.38 | 0.29 | 1.01 |
|  | SWC | 17.30 | 3.70 | 20.02 | 4.58 | 0.34 | 0.12 | 3.11 |
|  | NWC | 16.93 | 4.77 | 8.92 | 2.11 | 0.41 | 0.19 | −7.98 |
| **CO** | NEC | 0.64 | 0.17 | 0.38 | 0.12 | 0.48 | 0.61 | −0.27 |
| **(ppm)** | NC | 0.90 | 0.25 | 0.47 | 0.13 | 0.47 | 0.41 | −0.42 |
|  | CC | 0.81 | 0.17 | 0.58 | 0.14 | 0.49 | 0.45 | −0.22 |
|  | EC | 0.66 | 0.16 | 0.56 | 0.14 | 0.63 | 0.54 | −0.09 |
|  | SC | 0.66 | 0.11 | 0.32 | 0.08 | 0.36 | 0.41 | −0.34 |
|  | SWC | 0.69 | 0.14 | 0.49 | 0.11 | 0.49 | 0.29 | −0.19 |
|  | NWC | 0.89 | 0.25 | 0.25 | 0.04 | 0.35 | 0.18 | −0.63 |

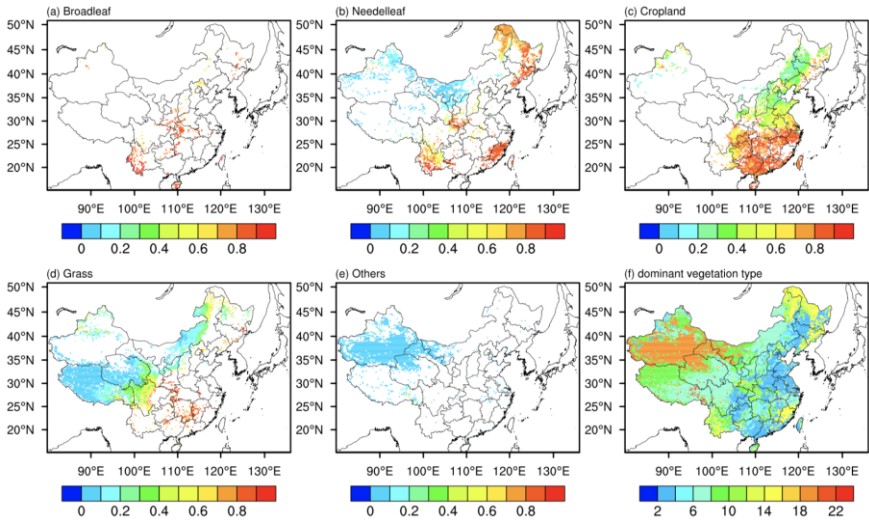

**Figure 1.** The vegetation fraction of (a) broadleaf, (b) needleleaf, (c) cropland, (d) grass, (e) others, and (f) dominant vegetation types.


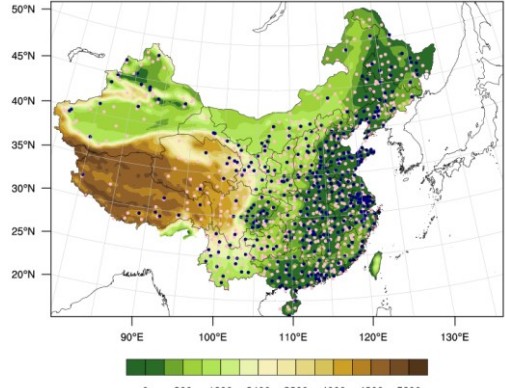

**Figure 2.** Site locations of air quality monitoring sites (blue dots) and the meteorological monitoring sites (pink dots) with the underlying is the terrain height (m).


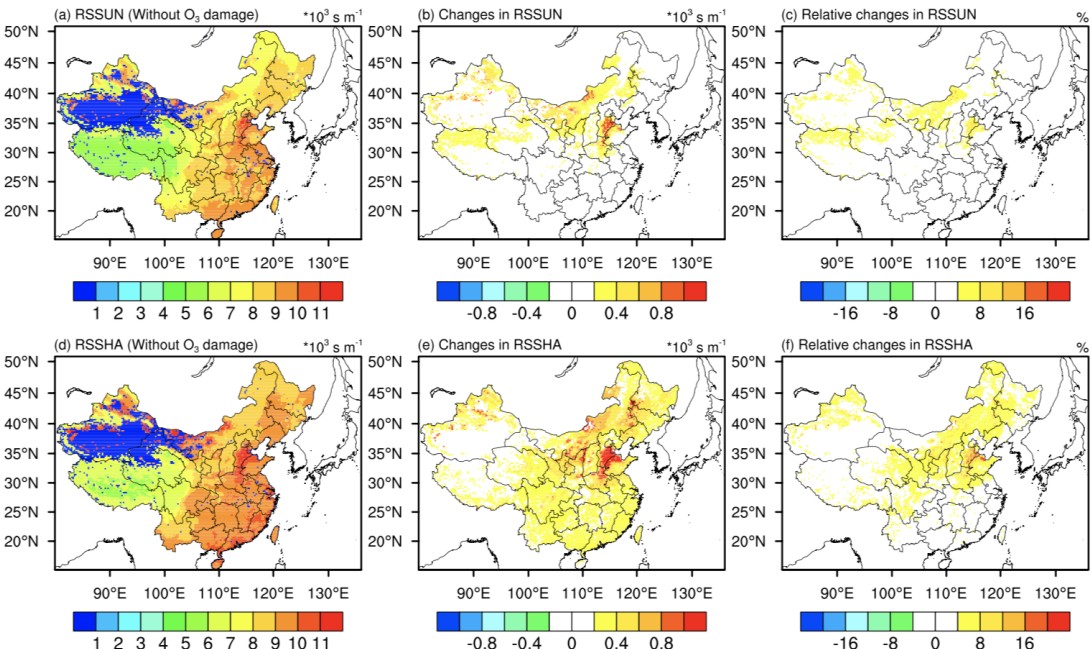

**Figure 3.** Spatial distribution of mean stomatal resistance in JJA of 2014–2017 for **(a)** sunlit leaves (RSSUN) and **(d)** shaded leaves (RSSHA) from the simu_withoutO$_3$ experiment. Absolute changes in **(b)** RSSUN and **(e)** RSSHA caused by O$_3$ damage. Relative changes in **(c)** RSSUN and **(f)** RSSHA caused by O$_3$ damage. Absolute changes are the RSSUN (RSSHA) from simu_withO$_3$ minus RSSUN (RSSHA) from simu_withoutO$_3$. Relative changes are calculated by absolute changes over the RSSUN (RSSHA) from simu_withoutO$_3$.

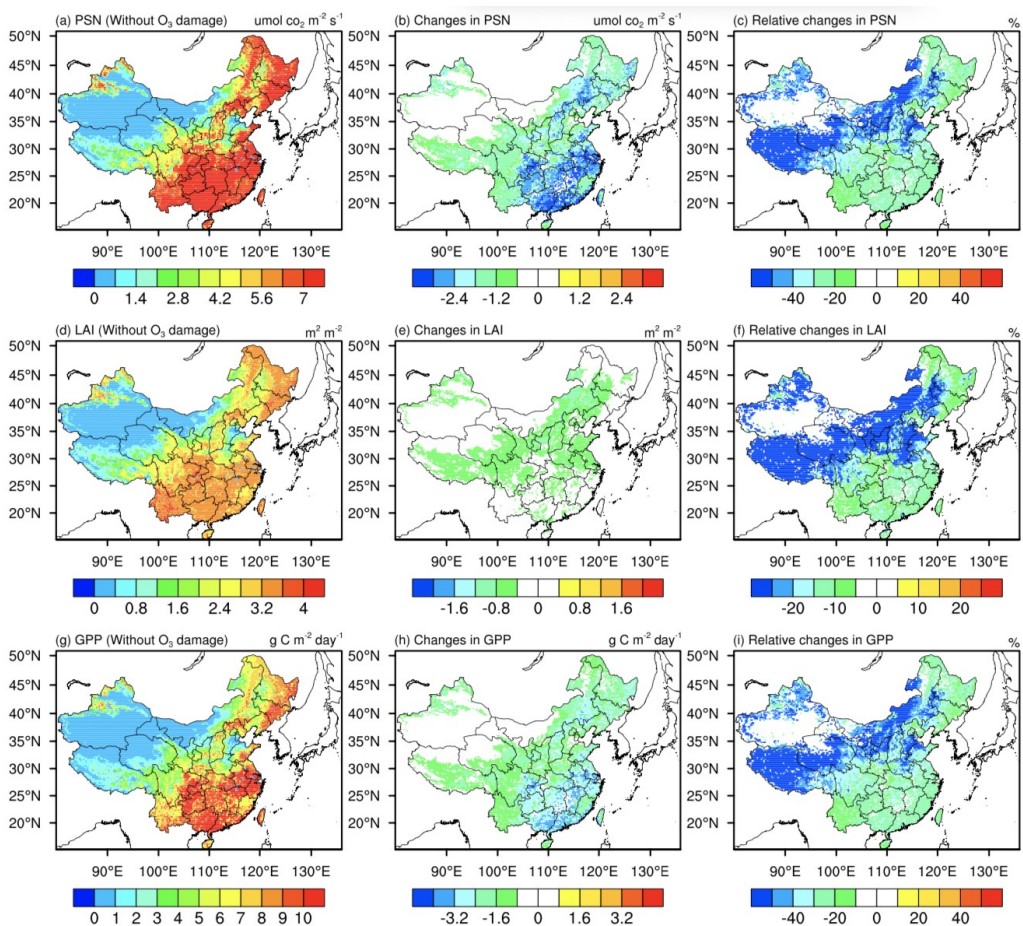

**Figure 4.** Spatial distribution of 2014–2017 JJA mean **(a)** photosynthesis rate (PSN), **(d)** leaf area index (LAI), and **(g)** gross primary productivity (GPP) from the simu_withoutO$_3$ experiment; absolute changes in **(b)** PSN, **(e)** LAI and **(h)** GPP caused by O$_3$ damage; and relative changes in **(c)** PSN, **(f)** LAI and **(i)** GPP caused by O$_3$ damage. Absolute changes are the results from simu_withO$_3$ minus results from simu_withoutO$_3$. Relative changes are calculated from the absolute changes over the results from simu_withoutO$_3$.




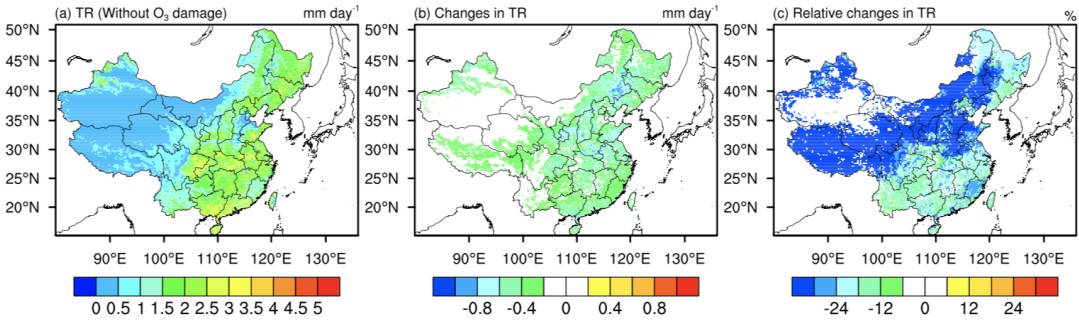

**Figure 5.** Spatial distribution of 2014–2017 JJA mean **(a)** transpiration rate (TR), and **(b)** absolute changes and **(c)** relative changes in TR caused by O$_3$ damage.

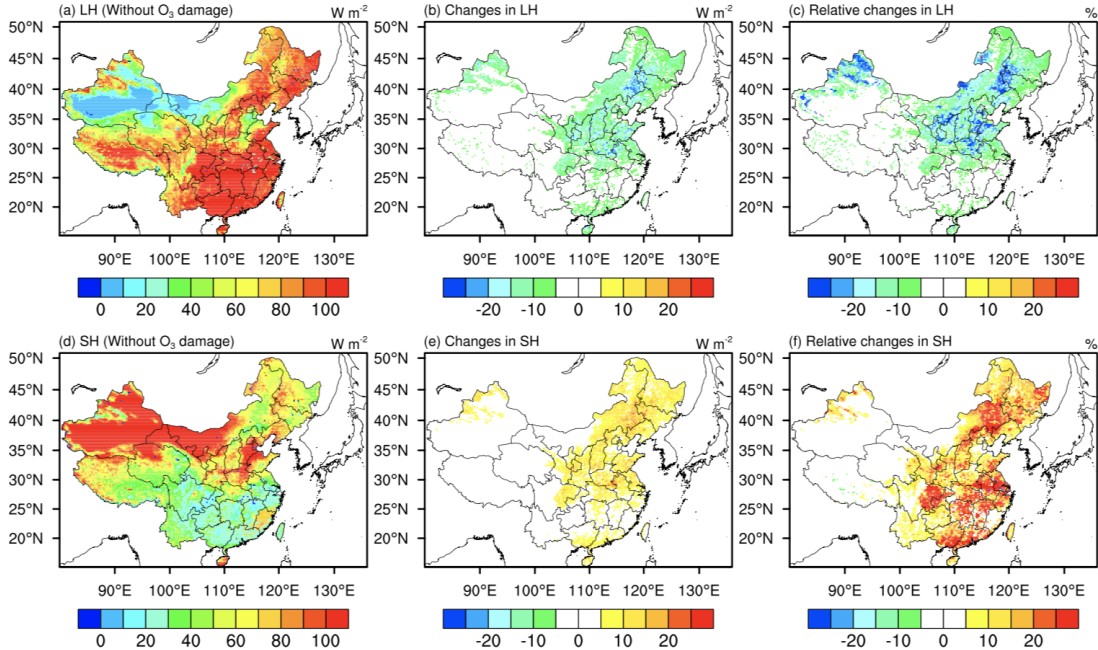

**Figure 6.** Spatial distribution of mean **(a)** latent heat flux (LH) and **(d)** sensitive heat flux (SH) from the simu_withoutO$_3$ experiment; absolute changes in **(b)** LH flux and **(e)** SH flux in JJA of 2014–2017 caused by O$_3$ damage; and relative changes in **(c)** LH flux and **(f)** SH flux caused by O$_3$ damage. Absolute changes are the LH (SH) flux from simu_withO$_3$ minus LH (SH) flux simu_withoutO$_3$. Relative changes are calculated by absolute changes over LH (SH) flux from simu_withoutO$_3$.

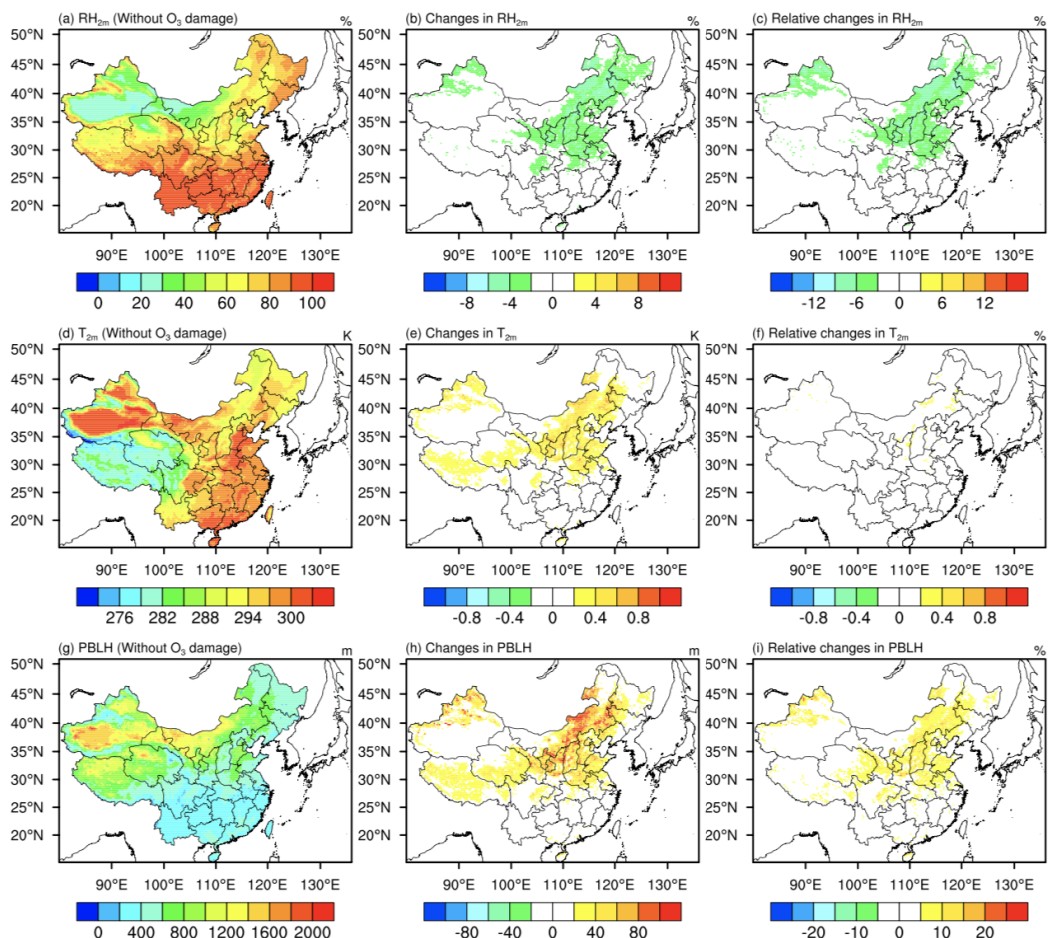

**Figure 7.** Spatial distribution of mean **(a)** 2-m relative humidity, **(d)** 2-m temperature at, and **(g)** planetary boundary layer height (PBLH) in JJA of 2014–2017 from the simu_withoutO$_3$ experiment; absolute changes in **(b)** RH$_{2m}$, **(e)** $T_{2m}$ and **(h)** PBLH caused by O$_3$ damage; and relative changes in **(c)** RH$_{2m}$, **(f)** $T_{2m}$ and **(i)** PBLH caused by O$_3$ damage. Absolute changes are the results from simu_withO$_3$ minus results from simu_withoutO$_3$. Relative changes are calculated by absolute changes over the results from simu_withoutO$_3$.

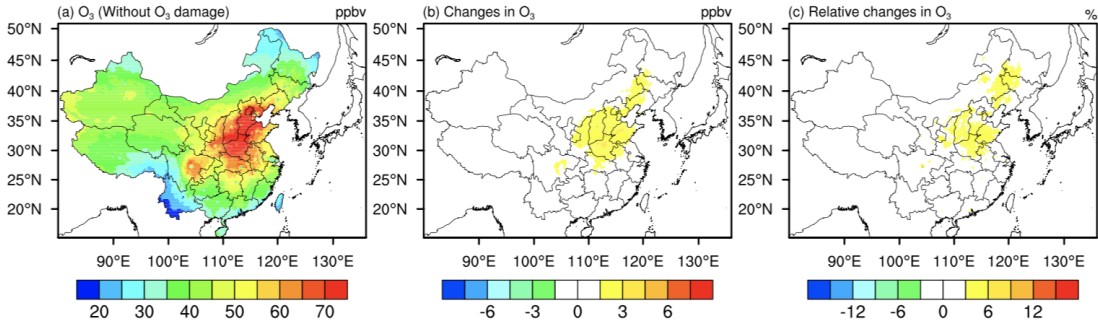

**Figure 8.** Same as Fig 5 but for surface O$_3$ concentration.

915

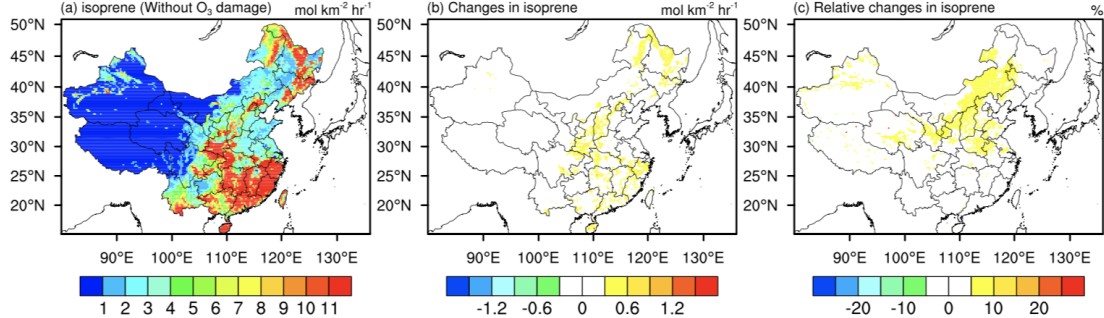

**Figure 9.** Same as Fig 5 but for isoprene emission.

920