# Peer review of "Effects of ozone-vegetation interactions on meteorology and air quality in China using a two-way coupled landatmosphere model"

_Atmospheric Chemistry and Physics, 2021_

## Referee Comment (RC1)

**Review ACP-2021-165**

Effects of ozone-vegetation interactions on meteorology and air quality in China using a two-way coupled land-atmosphere model

**1 general comments**

`evaluating the overall quality of the discussion paper`

- The authors study the effects of ozone damage on vegetation, air quality, and temperature (among others) in China with regional model simulations at 27 km horizontal resolution.

- The manuscript comprises results from model simulations with a *"revised version of the WRF-chem model"* (based on version 3.8.1) limited to 4 growing seasons (2014–2017).

- The authors use (implemented?) a two-way coupling between the atmospheric chemistry module (WRF-chem) and the land module (Noah-MP).

- The ozone damage scheme is based on the work of [1] (decoupled stomatal conductance and photosynthesis under cumulative ozone uptake).

- The used WRF-chem model setup (among myriads of reasonable choices) is listed in a comprehensive manner.

- Overall, the manuscript is decently written. The language is concise and comprehensible but needs refinement in terms of proper grammar (see an incomplete listing in Section 3).

- The results are presented in a comprehensible manner but more discussion is needed.

- Regarding the overall quality of the method. Large biases are reported between modeled concentrations of various species and observation. This is neither discussed nor are implications on the later results given. Hence, there are two major concerns (presented in more detail in Section 2):

  1. The large model biases are not discussed but made light of. Regardlessly, averages over all stations are computed. It would improve the point of the paper, if only those stations were taken into consideration for further analysis, which show a low model bias.

  2. The authors will not be able to fix the underlying issues in the model, but they should at least put an effort into discussion some of the reasons for the huge discrepancy between model and observation - maybe there are even quality issues with observations (representativeness)?

- The authors point out that they use a *"revised WRF-Chem model"* but are missing a section about **code availability** completely. Information on this has to be added.

- A section about **author contributions** seems to be missing, too.

- A section about **competing interests** is also missing (see manuscript composition on `https://www.atmospheric-chemistry-and-physics.net/submission.html#templates`).

**2 specific comments**

`individual scientific questions/issues`

- Section 2.3
  - L264–268: *"[...] each simulation was conducted from 24 May to 1 September [...] the days in May were discarded as spin-up [...] active growing season of plants."* Considering that ozone damage in the used formulation [1] is accumulative, the described method above potentially results in an underestimation of ozone uptake (before May 24), in particular for evergreen natural and semi-natural vegetation which is expected to be active even before May 24. The "active growing season" the authors refer to is for of crops, perhaps? How did the authors account for this potentially unaccounted ozone uptake? At least, it is highly recommended that they discuss this matter.
  - L270–282: *"Atmospheric forcing 2014–2017; anthropogenic emission 2014; [...] these years were selected based on high $O_3$ concentrations pointed out in previous studies [...]. [...] evaluated using available in-situ observations in China. [...] mean biases (MB), and correlation coefficients (CORR) [...]"* There is a contradiction between the chosen method and the intention described in the text. I assume anthropogenic emissions have been held constant at 2014 levels in these simulations. This is not properly expressed. Furthermore, computing the average divergence between model and observations in this case is not reflecting the model bias (as described by the authors). Given you'd have a perfect model and perfect anthropogenic precursor emissions for 2014, you would still expect larger divergence between model and observation for any other meteorological year than 2014. This is actually the case, as can be seen in Table 3 (MB values). Hence, the authors actually study the impact of different meteorological conditions (2014–2017) on pollutant concentrations in 2014. The authors should elaborate on this and rewrite these parts of their manuscript accordingly.
  - L319–330: *"The results indicate general overestimation by the model of most air pollutants except for CO. [...] but the spatial distribution of both meteorological variables and air pollutant concentrations are reasonably simulated by*

*the model, lending credence to the use of the model for sensitivity studies [...]"* CO is highly underestimated (Table 4). The bias in ozone concentrations in 2014 between the default model version and observations is of the same size as the observed concentrations. In conjunction with the large underestimations in CO concentrations, this may point to issues with either the ozone chemistry (to low titration perhaps) or a generally too low dry deposition in the model. In particular, the latter is affected by the implemented two-way coupling between ozone-induced damage on vegetation and the atmosphere and thus the main subject of this manuscript. The authors need to elaborate more on this and properly discuss reasons for the divergences (systematic uncertainties in both model and observations) and implications on the results.

- L350–352: *"Comparing the changes in RSSUN and RSSHA, [...], reflecting the larger sensitivity of shaded leaves to $O_3$ damage."* The authors should perhaps cite relevant articles, e.g. [2].

- L372–380: *"[...] where original PSN values are small [...]"* This seems to be mainly the case for arid regions in western China where the main vegetation type is grasslands. The text would benefit from referring not only to regions but also the associated types of vegetation in this regard.

- Section 3.3
  - L474–477: *"[...] we resort to use the more universal $O_3$ threshold of 40 ppb [...]"* The authors are referring to the ATO40 metric used for risk assessment of potential ozone damage on plants. ATO40 is an exposure-based metric not taking the actual uptake of ozone by the vegetation into account. In the context of this work, the actual damage is modeled and has been quantified as reduction in NPP/GPP. In L234, the **flux threshold** is given as $0.8 \, \text{nmol} \, O_3 \, \text{m}^{-2}$. Something does not add up in this paragraph, since these two thresholds are probably not interchangeable. I assume, that the authors are trying to say that previous studies used the AOT20/40 metric to assess potential ozone damage indirectly from modeled ozone concentrations. In their study they are able to directly assess the impact. And they consider the chosen limit on the flux (detoxification) as more conservative than studying AOT20. Then it is not the AOT threshold which is affecting the meteorology but the flux-threshold (L234). The authors should elaborate on this paragraph.

- Section 3.4
  - L497–499: *"$O_3$ concentrations increase the most (by up to 6 %) [...] with the maximum increment of 6 ppb."* Considering a model bias in ozone of the order of $100\%$ (biases of the same size as observation), this is not significant and should be clearly stated.

**3 technical corrections**

purely technical corrections

- L5-12: *Author affiliations* The affiliation indicated with * is missing from the list of affiliations.

- L74: *"Noah-MultiParamaterization"* Missing space and typo → *parameterization.*

- L75: *"CL M"* Remove space.

- L76: *"[...] is commonly used in to simulate [...]"* Remove *in.*

- L129: *"[...]A comprehensive study of how* $O_3$ *affects meteorology and air quality [...] is still limited but highly warranted."* This sentence is slightly unclear. Particularly, *warranted* might not be the right term in the context. The authors may consider revising it.

- L153: *"[...] and cover the whole China"* Grammar is probably off. Remove article?

- L161: *"[...] and an hourly resolution that were suitable [...]"* Consider rephrasing slightly: [...] and an 1 hourly resolution suitable [...].

- L168: *"with Secondary Organic Aerosol Model"* Probably needs an article (the) here.

- L294: *"[...] in year 2017 [...] in year 2014"* Remove word "year", respectively.

- L302–303: *"Fore example, the larger values [...]"* This sentence appears to be incomplete – it misses at least a verb. Please correct.

- L325: *"[...] at similar magnitude [...]"* Change preposition: *at* → *of.*

- L329: *"credence"* This term sounds odd in this context. The authors may consider rephrasing the sentence and use "trust" instead.

- L349: *"units break into new line"* Ought to be fixed here and other places in the following. Probably subject to final typesetting process, though.

- L473: *"[...] used in other previous studies[...]"* The authors should consider using either *other* or *previous.*

- L573: *"In this study, we found in China [...]"* This sentence is hard to read, the authors should elaborate on it. Maybe: *"In this study, we found that reduced dry deposition in China is mainly due to enhanced stomatal conductance, while enhanced isoprene emissions are mainly due to enhanced surface temperature and the corresponding increase in* $O_3$ *concentration.".*

- L584–605: *Grammar and sentence structure* is slightly off in this whole section and need refinement, in particular in L587 and L595.

- Page breaks: Some tables and figure captions are spread over several pages. Though, subject to final typesetting, this is slightly unpleasant. The authors may check their future manuscripts in this regards before submission.

- Diverging color bars:
The two colors associated with the highest negative divergence are too similar (not distinguishable on printout). The authors may consider fixing this.

**References**

[1] Lombardozzi, Danica and Levis, Samuel and Bonan, G. and Hess, P. and Sparks, Jed, *Temperature acclimation of photosynthesis and respiration: A key uncertainty in the carbon cycle-climate feedback*, J. Climate, vol. 28, pp. 292–305, 2015, doi: 10.1175/JCLI-D-14-00223.1

[2] Yoshiyuki Kinose and Yoshinobu Fukamachi and Shigeaki Okabe and Hiroka Hiroshima and Makoto Watanabe and Takeshi Izuta, *Photosynthetic responses to ozone of upper and lower canopy leaves of Fagus crenata Blume seedlings grown under different soil nutrient conditions*, Environ. Pollut., vol. 223, pp. 213–222, 2017, doi: 10.1016/j.envpol.2017.01.014

---

## Author Comment (AC1)

Response to Referee #1

**1 general comments**

evaluating the overall quality of the discussion paper

- The authors study the effects of ozone damage on vegetation, air quality, and temperature (among others) in China with regional model simulations at 27km horizontal resolution.
- The manuscript comprises results from model simulations with a "revised version of the WRF-chem model" (based on version 3.8.1) limited to 4 growing seasons (2014−2017).
- The authors use (implemented?) a two-way coupling between the atmospheric chemistry module (WRF-chem) and the land module (Noah-MP).
- The ozone damage scheme is based on the work of [1] (decoupled stomatal conductance and photosynthesis under cumulative ozone uptake).
- The used WRF-chem model setup (among myriads of reasonable choices) is listed in a comprehensive manner.
- Overall, the manuscript is decently written. The language is concise and comprehensible but needs refinement in terms of proper grammar (see an incomplete listing in Section 3).
- The results are presented in a comprehensible manner but more discussion is needed.

Response: We thank the referee for the positive comments and valuable suggestions that are helpful for improving our manuscript. All the questions and concerns raised have been carefully discussed and addressed. We have provided a point-by-point response to the reviewers' comments below in blue color.

- Regarding the overall quality of the method. Large biases are reported between modeled concentrations of various species and observation. This is neither discussed nor are implications on the later results given. Hence, there are two major concerns (presented in more detail in Section 2):
1. The large model biases are not discussed but made light of. Regardlessly, averages over all stations are computed. It would improve the point of the paper, if only those stations were taken into consideration for further analysis, which show a low model bias.
2. The authors will not be able to fix the underlying issues in the model, but they should at least put an effort into discussion some of the reasons for the huge discrepancy between model and observation - maybe there are even quality issues with observations (representativeness)?

Response: Thanks for your constructive suggestions. Yes, we agree that averaging the results of all stations may cause more biases. A more detailed model evaluation and discussion talking about the discrepancies have been conducted and now included in the revised manuscript (Line 303–309, Line 311–332) and in the supplement (Tables S4–S10, Tables S12–S23).

- The authors point out that they use a "revised WRF-Chem model" but are missing a section about **code availability** completely. Information on this has to be added.
  Response: This section is now added in the revised manuscript.
- A section about author contributions seems to be missing, too.
  Response: Added with thanks.
- A section about competing interests is also missing (see manuscript composition on https://www.atmospheric-chemistry-and-physics.net/submission.html#templates).
  Response: Added with thanks.

**2 specific comments**

individual scientific questions/issues

- Section 2.3

L264−268: "[...] each simulation was conducted from 24 May to 1 September [...] the days in May were discarded as spin-up [...] active growing season of plants." Considering that ozone damage in the used formulation [1] is accumulative, the described method above potentially results in an underestimation of ozone uptake (before May 24), in particular for evergreen natural and seminatural vegetation which is expected to be active even before May 24. The "active growing season" the authors refer to is for of crops, perhaps? How did the authors account for this potentially unaccounted ozone uptake? At least, it is highly recommended that they discuss this matter.

Response: Thank you for the comment. It is true that the simulation period of our study could not cover all the periods when $O_3$ damage happens and may not cover the growing season of all vegetation types, which may cause uncertainties. We have revised the sentence in Section 2.3 to make this point clearer. The sentence in Line 276–277 in Section 2.3 has been modified as follows: "JJA was selected because of the most severe $O_3$ pollution in this season and because it is within the active growing season of the plants."

We have also acknowledged the uncertainty caused by this problem in the conclusion section in Line 543–549 as follows: "In this study, the summertime simulation period of JJA was selected due to the high O$_3$ pollution in this season and the overlapping with vegetation growing season to capture the severe O$_3$ damage on vegetation. Nevertheless, uncertainty may still arise from that our simulation period may not cover the growing season of all vegetation types and may not cover all periods that O$_3$ damage happens, which may represent an underestimation of the full scale of O$_3$ damage. Future work should be conducted for longer time periods and for all seasons, which will help us better understand O$_3$-vegetation interactions in China."

L270−282: "Atmospheric forcing 2014−2017; anthropogenic emission 2014; [...] these years were selected based on high O$_3$ concentrations pointed out in previous studies [...]. [...] evaluated using available in-situ observations in China. [...] mean biases (MB), and correlation coefficients (CORR) [...]"
There is a contradiction between the chosen method and the intention described in the text. I assume anthropogenic emissions have been held constant at 2014 levels in these simulations. This is not properly expressed.
Furthermore, computing the average divergence between model and observations in this case is not reflecting the model bias (as described by the authors). Given you'd have a perfect model and perfect anthropogenic precursor emissions for 2014, you would still expect larger divergence between model and observation for any other meteorological year than 2014. This is actually the case, as can be seen in Table 3 (MB values). Hence, the authors actually study the impact of different meteorological conditions (2014−2017) on pollutant concentrations in 2014. The authors should elaborate on this and rewrite these parts of their manuscript accordingly.
Response: Thanks for this comment. This part has been revised in Section 2.3 in Line 274–275 as follows: "For each simulation in the four years, anthropogenic emissions were kept at 2014 levels, while meteorological fields were changing every year."
This sentence has been deleted: "These years were selected based on the high O$_3$ concentrations that were pointed out in previous studies (Li et al., 2018; Lu et al., 2018; Silver et al., 2018)."

It is true that the changes in air pollutants are mainly driven by changes in meteorological fields since anthropogenic emissions are kept in 2014 levels. We have tried our best to use the most updated anthropogenic emission inventory that we have in our simulations. We would like to clarify that this study did not aim to investigate the role of emissions or the role of meteorology on air pollutants. The main scope of this study is to investigate the effect of O$_3$ damage on vegetation and the following changes in meteorology and feedbacks on O$_3$ concentration. We recognize this limitation could cause uncertainties and it should be mentioned in the paper, so we added the following in Line 566–568 in the discussion section as follows: "It should also be noted that keeping the anthropogenic emission inventory fixed in 2014 levels may be another limitation because of the nonlinear chemistry involving biogenic and anthropogenic precursors."

L319−330: "The results indicate general overestimation by the model of most air pollutants except for CO. [...] but the spatial distribution of both meteorological variables and air pollutant concentrations are reasonably simulated by the model, lending credence to the use of the model for sensitivity studies [...]"
CO is highly underestimated (Table 4). The bias in ozone concentrations in 2014 between the default model version and observations is of the same size as the observed concentrations. In conjunction with the large underestimations in CO concentrations, this may point to issues with either the ozone chemistry (to low titration perhaps) or a generally too low dry deposition in the model. In particular, the latter is affected by the implemented two-way coupling between ozone-induced damage on vegetation and the atmosphere and thus the main subject of this manuscript. The authors need to elaborate more on this and properly discuss reasons for the divergences (systematic uncertainties in both model and observations) and implications on the results.
Response: Thank you for this comment. We agree that more explanation should be included in the revised manuscript. The explanation is now added in Line 316–318 in the revised manuscript as follows: "The underestimation of CO can be explained by either O$_3$ chemistry, which points to the problem related to low titration, or in the underestimation of dry deposition by the model, which is also affected by the modification of the model."

L350−352: "Comparing the changes in RSSUN and RSSHA, [...], reflecting the larger sensitivity of shaded leaves to O$_3$ damage." The authors should perhaps cite relevant articles, e.g. [2].
Response: Thanks for the references, which are now included in the revised manuscript in Line 353.

L372−380: "[...] where original PSN values are small [...]" This seems to be mainly the case for arid regions in western

China where the main vegetation type is grasslands. The text would benefit from referring not only to regions but also the associated types of vegetation in this regard.

Response: Thanks for this comment. The text has been revised in Line 367–369 in Section 3.2 in the revised manuscript as follows: **"**In western China where the dominant vegetation type is grassland and the original PSN values are small, more than 40% of PSN is reduced due to $O_3$ damage (Fig. 3c).**"**

- Section 3.3

L474−477: "[...] we resort to use the more universal $O_3$ threshold of 40 ppb [...]"

The authors are referring to the ATO40 metric used for risk assessment of potential ozone damage on plants. ATO40 is an exposure-based metric not taking the actual uptake of ozone by the vegetation into account. In the context of this work, the actual damage is modeled and has been quantified as reduction in NPP/GPP. In L234, the flux threshold is given as 0.8 nmol$O_3$ m$^{-2}$. Something does not add up in this paragraph, since these two thresholds are probably not interchangeable. I assume, that the authors are trying to say that previous studies used the AOT20/40 metric to assess potential ozone damage indirectly from modeled ozone concentrations. In their study they are able to directly assess the impact. And they consider the chosen limit on the flux (detoxification) as more conservative than studying AOT20. Then it is not the AOT threshold which is affecting the meteorology but the flux-threshold (L234). The authors should elaborate on this paragraph.

Response: Thanks for this comment. The paragraph has been rewritten in Line 438–444 in Section 3.3 in the revised manuscript as follows: "However, in their study, Li et al. (2016) assumed that $O_3$ damage to plants happens when $O_3$ concentration is over a threshold of 20 ppb to imitate a weaker detoxifying effect of plants, instead of the 40 ppb threshold that was commonly used in previous studies. Considering the severe $O_3$ air pollution in China, we resorted to use the more universal $O_3$ threshold used by previous studies (Lombardozzi et al., 2015; Sadiq et al., 2017; Zhou et al., 2018) to represent a more conventional detoxifying effect, instead of lowering the threshold value that would cause much larger changes in the surface fluxes and meteorological fields."

- Section 3.4

L497−499: "$O_3$ concentrations increase the most (by up to 6 %) [...] with the maximum increment of 6 ppb." Considering a model bias in ozone of the order of 100% (biases of the same size as observation), this is not significant and should be clearly stated.

Response: Thanks for this comment. This has been included in Line 473–475 in the revised manuscript as follows: "It should be cautiously noted that in terms of magnitude alone the model biases in $O_3$ are comparable and sometimes larger than the up to 6 ppb systematic enhancement caused by $O_3$ damage, which represents be one major source of uncertainties in our study."

**3 technical corrections**

purely technical corrections

- L5-12: Author affiliations. The affiliation indicated with * is missing from the list of affiliations.
  Response: Modified with thanks.
- L74: "Noah-MultiParamaterization" Missing space and typo-> parameterization.
  Response: Modified with thanks.
- L75: "CL M" Remove space.
  Response: Modified with thanks.
- L76: "[...] is commonly used in to simulate [...]" Remove in.
  Response: Removed in the revised manuscript.
- L129: "[...]A comprehensive study of how $O_3$ affects meteorology and air quality [...] is still limited but highly warranted." This sentence is slightly unclear. Particularly, warranted might not be the right term in the context. The authors may consider revising it.
  Response: The sentence has been revised in Line 133–137 as follows: "However, a comprehensive study of how $O_3$ affects meteorology and air quality through $O_3$-vegetation interactions in China at high spatial resolutions, especially under severe $O_3$ pollution, is still limited but highly needed. Moreover, there have been limited studies focusing on the feedbacks of $O_3$-vegetation coupling on $O_3$ concentration itself, especially in China, which is one of the main scopes of our study."
- L153: "[...] and cover the whole China" Grammar is probably o_. Remove article?
  Response: Removed with thanks.
- L161: "[...] and an hourly resolution that were suitable [...]" Consider rephrasing slightly: [...] and an 1 hourly resolution suitable [...].

Response: Revised with thanks.

- L168: "with Secondary Organic Aerosol Model" Probably needs an article (the) here.
  Response: Added with thanks.
- L294: "[...] in year 2017 [...] in year 2014" Remove word "year", respectively.
  Response: Removed with thanks.
- L302−303: "For example, the larger values [...]" This sentence appears to be incomplete - it misses at least a verb. Please correct.
  Response: Corrected in the revised manuscript.
- L325: "[...] at similar magnitude [...]" Change preposition: at -> of.
  Response: Modified with thanks.
- L329: "credence" This term sounds odd in this context. The authors may consider rephrasing the sentence and use "trust" instead.
  Response: Modified with thanks.
- L349: "units break into new line" Ought to be fixed here and other places in the following. Probably subject to final typesetting process, though.
  Response: Fixed in the revised manuscript.
- L473: "[...] used in other previous studies[...]" The authors should consider using either other or previous.
  Response: Modified with thanks.
- L573: "In this study, we found in China [...]" This sentence is hard to read, the authors should elaborate on it. Maybe: "In this study, we found that reduced dry deposition in China is mainly due to enhanced stomatal conductance, while enhanced isoprene emissions are mainly due to enhanced surface temperature and the corresponding increase in $O_3$ concentration.".
  Response: Modified as suggested.
- L584−605: Grammar and sentence structure is slightly off in this whole section and need refinement, in particular in L587 and L595.
  Response: The paragraph has been rewritten in Line 545–572 in the revised manuscript as follows: "Nevertheless, uncertainty may still arise from that our simulation period may not cover the growing season of all vegetation types and may not cover all periods that $O_3$ damage happens, which may represent an underestimation of the full scale of $O_3$ damage. Future work should be conducted for longer time periods and for all seasons, which will help us better understand $O_3$-vegetation interactions in China. Uncertainty may also arise from the $O_3$ scheme employed in this study in terms of the CUO calculation and the consideration of $O_3$ detoxification mechanism of different vegetation types. The calculation of CUO heavily relies on the $O_3$ threshold. Considering the sensitivities of different vegetation types to $O_3$ damage, CUO threshold should be varied with different vegetation types. However, a constant $O_3$ threshold was employed in our study for the whole simulation domain and for all vegetation types, which may either underestimate or overestimate the actual $O_3$ damage. Moreover, following the work of Lombardozzi et al. (2015), we classified all the vegetation types into only three groups, which may be too coarse to investigate $O_3$ damage effects on regional or local scales. For example, Zhou et al. (2018) pointed out that Lombardozzi et al. (2015) treated tropical and temperate plants equivalently, which might lead to possible biases. More studies should be conducted to derive more appropriate $O_3$ thresholds for CUO calculation and make them available for regional scales or for different vegetation types. Another source of uncertainty may arise from the lack of representation of the direct effect of $O_3$ on isoprene emission. As pointed out by Gong et al. (2020), including the effect of $O_3$ damage on isoprene emission may reduce $O_3$ concentration by influencing precursors, but increase $O_3$ concentration at the same time through weakening the shortwave radiative forcing of secondary organic aerosols, which would help constitute a more complete feedback mechanism between $O_3$ and vegetation. Moreover, uncertainties may also come from that the effect of soil moisture deficit was not considered in this study, which may underestimate the reduction in dry deposition sink of $O_3$. It should also be noted that keeping the anthropogenic emission inventory fixed in 2014 levels may be another limitation because of the nonlinear chemistry involving biogenic and anthropogenic precursors. Despite these uncertainties and limitations, our study provides detailed and comprehensive results whereby $O_3$-vegetation impacts will adversely affect plant growth and crop production, contribute to global warming, worsen the severe $O_3$ air pollution in China via feedbacks, and identifies the hotspot areas in the country. Our findings clearly pinpoint the need to consider the $O_3$ damage effects in both air quality studies and climate change studies."

- Page breaks: Some tables and figure captions are spread over several pages. Though, subject to final typesetting, this is slightly unpleasant. The authors may check their future manuscripts in this regard before submission.

Response: Thanks for this comment. The problem has been fixed in the revised manuscript.

- Diverging color bars:
  The two colors associated with the highest negative divergence are too similar (not distinguishable on printout). The authors may consider fixing this.
  Response: Thanks for pointing out this problem. Figures have been modified in the revised manuscript.

References

[1] Lombardozzi, Danica and Levis, Samuel and Bonan, G. and Hess, P. and Sparks, Jed, Temperature acclimation of photosynthesis and respiration: A key uncertainty in the carbon cycle-climate feedback, J. Climate, vol. 28, pp. 292−305, 2015, doi: 10.1175/JCLI-D-14-00223.1

[2] Yoshiyuki Kinose and Yoshinobu Fukamachi and Shigeaki Okabe and Hiroka Hiroshima and MakotoWatanabe and Takeshi Izuta, Photosynthetic responses to ozone of upper and lower canopy leaves of Fagus crenata Blume seedlings grown under different soil nutrient conditions, Environ. Pollut., vol. 223, pp. 213−222, 2017, doi:10.1016/j.envpol.2017.01.014

---

## Author Comment (AC2)

Response to Referee #2

We thank the referee for the thoughtful review and the constructive suggestions that are helpful to improve our manuscript. All the questions and concerns raised have been carefully discussed and answered. Below shown in blue color is the point-by-point response to the referee's comments.

Major comments:

1.  The study use a regional chemical transport model (WRF-Chem) with a revised ozone-damage scheme to explore the sensitivity of meteorology and ozone air quality to ozone-vegetation interactions, specifically, ozone damage. The authors discussed that most of the model sensitivity results presented in this study are broadly consistent with the results from the earlier studies (e.g., Sadiq et al., 2017). It is not clear to the referee what the novelty of this specific research article is compared to the earlier studies. Furthermore, the ozone-vegetation interactions and meteorological responses discussed in this study appear to be purely based on model sensitivity experiments, thus missing critical observational constrains. The authors conducted some evaluation of surface meteorology, ozone and related chemical tracers averaged over entire China from their base (?) simulation, but there are no evaluation and discussion regarding how the introduction of ozone-vegetation interactions in the model improves the simulation of ozone air quality and surface meteorology. The model sensitivity results will be much more trustworthy if the authors could demonstrate that the new model with ozone-damage substantially improves the simulation of observed ozone interannual variability and mean distributions, at least over the areas where the ozone-vegetation interactions are largest.

Response: Thanks for the comments, which are highly appreciated. It is true that our results are generally consistent with previous studies, such as Sadiq et al. (2017), due to the same $O_3$ damage scheme following Lombardozzi et al. (2015) being employed. However, we would like to clarify that previous studies mainly focused on the global scale with coarse resolutions, which often failed to capture the spatial distribution of $O_3$ damage on vegetation in China. Based on the results from global studies pointing out that China is a hotspot in terms of $O_3$ pollution and $O_3$ damage on vegetation, our model simulations performed at higher spatial resolutions were capable of investigating $O_3$ damage effect on regional or provincial scales in China. Moreover, there have been limited studies focusing on the feedbacks of $O_3$-vegetation coupling on $O_3$ concentration itself, especially in China, which is one of the main scopes of our study. In addition, different from the work of Sadiq et al. (2017) that mainly examined feedbacks on $O_3$ concentration, our work also investigates the effects of $O_3$-vegetation interactions on boundary-layer meteorology in China. We now make the novelty of this study more clearly stated in the introduction and the discussion section of the revised manuscript.

"However, a comprehensive study of how $O_3$ affects meteorology and air quality through $O_3$-vegetation interactions in China at high spatial resolutions, especially under severe $O_3$ pollution, is still limited but highly needed. Moreover, there have been limited studies focusing on the feedbacks of $O_3$-vegetation coupling on $O_3$ concentration itself, especially in China, which is one of the main scopes of our study."

"Previous studies mainly focused on the global scale with coarse spatial resolutions, which did not fully capture the spatial distribution of $O_3$ damage on vegetation in China. Based on the results from global studies pointing out that China is a hotspot in terms of $O_3$ pollution and $O_3$ damage on vegetation, our model simulations performed at high spatial resolutions were capable of investigating $O_3$ damage effects on regional and provincial scales in China."

Yes, we agree that we should mention how the introduction of $O_3$ damage affects the performance of the model. The following sentences are now included in  in the manuscript as follows: "We also compared the evaluation results between the original model and the modified model, as shown in Table S2 and Table S3 in the supplement and Table 3 and Table 4 here. We found no obvious differences in the evaluation results between the original model results and the revised model results. It should be noted that this study might not be able to and was not meant to improve model accuracy, but our modified model is able to capture $O_3$-vegetation interactions without worsening model performance." In response to this comment, the evaluation results of the original model simulations are included in the supplement (Table S2 and Table S3). The typo error in Table 3 is also fixed.

2.  From Table 4, it appears that the model not only has large mean-state ozone biases and but also have difficulty simulating the observed ozone interannual variability. For example, observations are lowest in JJA 2014 and highest in JJA 2017. The model does not capture this variability at all. It is not clear from the text and table captions as to which model they are evaluating, the old model without ozone damage, or the new model with ozone damage? Does the new model with ozone damage better simulate the observed high-ozone summer and extreme events? If not, why shall we care all the sensitivity results discussed in the paper? Also in Table 4 and Fig.8, are you showing JJA average of 24-hour mean ozone or daily maximum 8 hour average ozone (MDA8)? Since the effects of ozone damage via stomatal uptakes are expected to be largest during daytime, the analysis should focus on daytime or MDA8 ozone, not the 24-hour average.

Response: Thank you for the comment. The evaluation results shown in the manuscript are from the new model. We recognize the limitation that our model may not be to capture the interannual variations in $O_3$ concentration. It may be attributable to the representation of anthropogenic emission inventory whereby anthropogenic emissions were kept at 2014 levels for all four years of simulations. Nevertheless, it should be clarified that we have employed the most updated emission inventory used in the model simulations. Even though the interannual variations may not be resolved, the results still show the systematic differences with and without $O_3$ damage, which is the main scope of our study. We agree with the referee and this limitation should be mentioned in the paper, so we added the following sentences in Line 566–568 in the discussion section as follows: "It should also be noted that keeping the anthropogenic emission inventory fixed in 2014 levels may be another limitation because of the nonlinear chemistry involving biogenic and anthropogenic precursors."

In terms of the performance of the new (modified) model and the old (default) model, as shown in Table S2 and Table S3 in the supplement and Table 3 and Table 4 in the manuscript, we found no significant differences in the evaluation results between the original model results and the new model results. Improving the model performance is beyond the scope our study. Moreover, it should be clarified that this study may not be able to improve model accuracy. Nevertheless, our modified model can capture two-way $O_3$-vegetation interactions without worsening the evaluation, unlike in the study of Sadiq et al. (2017).

The average of 24-hour $O_3$ is used in Table 4 and Figure 8. We agree that showing the changes in daytime is meaningful. The results for the changes in daytime $O_3$ are shown below. Similar distribution and changes caused by $O_3$ damage are found in Fig. S1 with those of Figure 8, we therefore still use Fig. 8 in the main text in consistency with other figures, which all show the results of average of the 24-hour. The results showing the daytime changes are included in the supplement in Figure S1.

[Figure]

**Figure S1.** Spatial distribution of 2014–2017 JJA daytime mean **(a)** surface $O_3$ concentration, and **(b)** absolute changes and **(c)** relative changes in $O_3$.

3.  From Figure 3, it appears that changes in vegetation properties due to ozone damage are most prominent in areas with sparse vegetation, such as north and northwest China. Why? The authors report the large percentage change in the abstract, but this could be misleading, as the large percentage change could be the numerical artifact from dividing a small value.

Response: Thank you for this comment. Yes, it is true that reductions in vegetation properties (PSN, GPP) are found over northern and northwestern China. Northern China is covered by croplands and needleleaf trees, while northwestern China is covered by grasses and needleleaf trees, which are both sensitive to $O_3$ damage.

We agree that using percentage change that could be misleading. We have revised in Line 27–30 in the abstract to avoid it as follows: "$O_3$ damage causes more than 0.6 μmol $CO_2$ $m^{-2}$ $s^{-1}$ reductions in photosynthesis rate, and at least 0.4 and 0.8 g C $m^{-2}$ $day^{-1}$ decrease in leaf area index (LAI) and gross primary production (GPP), respectively, and hotspot areas appear in the northeastern and southern China."

Other comments:

1. Lines 50-70 and 95-115: there are a few recent papers demonstrating the significant impacts of reduced ozone removal by drought-stressed vegetation on observed surface ozone trends and extremes. These papers can be discussed here for a complete literature review:

Huang, L., McDonald-Buller, E. C., McGaughey, G., Kimura, Y. & Allen, D. T. The impact of drought on ozone dry deposition over eastern Texas. Atmos. Environ. 127, 176–186 (2016).

Lin, M. et al. Sensitivity of ozone dry deposition to ecosystem–atmosphere interactions: a critical appraisal of observations and simulations. Glob. Biogeochem. Cycles 33, 1264–1288 (2019).

Lin, M., Horowitz, L.W., Xie, Y. et al. Vegetation feedbacks during drought exacerbate ozone air pollution extremes in Europe. Nat. Clim. Chang. 10, 444–451 (2020). https://doi.org/10.1038/s41558-020-0743-y

Response: Thank you for the comment and the references. We agree that the impacts from drought-stress should be included in the introduction. Such impacts from drought stress have now been included in Line 65–67 in the introduction as follows: "Moreover, recent studies showed reduced dry deposition velocities of $O_3$ by drought-stressed vegetation, which affects surface $O_3$ trends and extremes (Huang et al., 2016; Lin et al., 2019; Lin et al., 2020)."

We also recognize the limitation of not considering the drought stress should be mentioned in the paper. The limitation of not considering the drought stress of our study is now acknowledged in Line 564–566

in Section 4 in the revised manuscript as follows: "Moreover, uncertainties may also come from that the effect of soil moisture deficit was not considered in this study, which may underestimate the reduction in dry deposition sink of $O_3$."

2. Lines 155-160, clarify you are using monthly mean chemical boundary conditions from MOZART?

Response: Thanks for this comment. Yes, we are using monthly mean chemical boundary conditions from MOZART. The sentence has been modified in Line 166–168 as follows: "The chemical initial and boundary conditions were generated from the Model for Ozone and Related Chemical Tracer version 4 (MOZART-4), which is available at a horizontal resolution of 1.9°×2.5° with 56 vertical layers (Emmons et al., 2010)."

1. Simulation years should be clarified in Section 2.1

Response: Thanks for the comment. The simulation years have been clarified in Line 157–158 in Section 2.1 as follows: "Simulations are conducted from 24 May to 1 September every year from 2014 to 2017 and the days in May were discarded as spin-up."

4. Section 2.2:

(1) This section should include information on the fraction of sunlit and shaded leaves as well as the fraction of dominant vegetation types considered in the model. Fig.4 fits better in this section.

Response: Thanks for this comment. The figure showing the distribution of vegetation fraction of dominant vegetation types in China has been moved to Section 2.2. The description is also added in Line 200–204 as follows: "The land use types and the vegetation parameters are based on the U.S. Geological Survey (USGS) embedded in Noah-MP. Fig. 1 shows the spatial distribution of vegetation fraction of dominant vegetation types in China. The distribution of main vegetation groups (broadleaf, needleleaf, crop and grass) that have different sensitivities to $O_3$ damage following Lombardozzi et al. (2015) are shown in Fig. 1."

We appreciate the reviewer's insightful suggestion and agree that it would be useful to show the fraction of sunlit and shaded leaves. However, the output of the fraction of sunlit and shaded leaves is available unless the modification of the model and rerunning the simulations, which is beyond the main scope of this study. To address this comment, the consideration of sunlit and shaded leaves in Noah-MP is added in Line 195–198 in Section 2.2 to help answer this question. "Noah-MP also considers the photosynthesis of sunlit and shaded leaves separately, whereby sunlit leaves are more limited by $CO_2$ concentration

while shaded leaves are more constrained by insolation, which may thus have different responses to $O_3$ damage."

(2) It is not clear from the text whether the authors implement a new ozone dry deposition and damage/feedback scheme in the WRF-Chem model. Does the simulated stomatal resistance respond to soil moisture deficits? According to several recent papers listed above, stomatal closure induced by soil moisture deficits can substantially increase surface ozone concentrations; this process is an important part of the ozone-vegetation interactions. The default Ball-Berry scheme does not include the effects of soil moisture. The default Wesely dry deposition scheme used in WRF-Chem does not consider the effects of soil moisture, neither (e.g., Rydssa et al., 2016).

Rydsaa, J. H., Stordal, F., Gerosa, G., Finco, A. & Hodnebrog, O. Evaluating stomatal ozone fluxes in WRF-Chem: comparing ozone uptake in Mediterranean ecosystems. Atmos. Environ. 143, 237–248 (2016). The role of soil moisture should be clearly discussed and clarified in the manuscript.

Response: Thanks for the valuable comment and the references. Yes, the default FBB model and Wesely model were employed in this study and the role of soil moisture deficit was not considered in this study. We agree with the referee that soil moisture deficit should be considered to help better understand the $O_3$-vegetation interactions. We have emphasized the role of soil moisture deficit in Line 468–470 in Section 3.4. The revised sentences are as follow: "Soil moisture deficit, which has been shown to reduce stomatal uptake, if considered, will also contribute to the enhancement in $O_3$ concentration (Rydsaa et al., 2016)."

We also acknowledged this limitation in Line 564–566 in Section 4 in the revised manuscript as follows: "Moreover, uncertainties may also come from that the effect of soil moisture deficit was not considered in this study, which may underestimate the reduction in dry deposition sink of $O_3$."

2. Tables 3 and 4. The evaluation should be done by the different parts of China, according to ozone pollution conditions, meteorological regimes, and vegetation types., and tied closely to the model sensitivity experiments, as discussed in my major comments.

Response: Thank you for the comment. We agree that showing evaluation results for different parts of China will be helpful. The evaluation results of 30 major cities in China are shown in the supplement. In response to the referee's comments, as suggested, the evaluation for seven major geographic regions of China have also been conducted. The results and discussion have been added in the revised manuscript and in the supplement.

---

## Author Response (AR2)

- A comprehensive analysis of major regions in China was included in the revision but kept in the supplement. These results should be promoted to the main manuscript (if possible in an abridged table or bar chart form) as the applicability of whole-China averages based on the model data was one of the major concerns in the previous review round.

Response: Thank you for the comment. Tables S8-S10 showing the evaluation results of meteorological variables have been replaced by Table 4 in the main manuscript. Tables S18-S23 showing the evaluation results of air pollutants have been replaced by Table 6 in the main manuscript.

- It is interesting that the two-way coupling neither improves (as one might hope for) nor worsens (as seemed to be the major concern by the authors) the model performance (L325-330). This somewhat contradicts the purpose of the manuscript (which is to point out the effects of two-way coupling on surface climate and air pollution estimates in China). Maybe some elaboration on the benefits / downsides of two-way coupling is needed in the introduction?

Response: Thank you for the comment. As pointed out in the main manuscript, the main purpose of this study is to investigate how $O_3$ affects meteorology and air quality through $O_3$-vegetation interactions and the feedbacks of $O_3$-vegetation coupling on $O_3$ concentration itself in China, which has not been examined before. For the model performance, we did not hope that the implemented model will help improve the model performance a lot at this stage considering the uncertainties that we mentioned in the discussion. We mainly intended to show that the implemented model used in this study can capture the $O_3$-vegetation interactions without the expense of worsening the model performance. With the consideration of soil moisture deficit, the detailed vegetation type classification, the more appropriate $O_3$ damage thresholds, we reasonably hope that model performance can be improved in the future, as mentioned in the discussion.

In this study, the land surface processes, atmospheric dynamics, and atmospheric chemistry in the WRF-Chem model were fully coupled. The two-way coupled model allows the $O_3$ concentration simulated by the chemical model to be dynamically passed onto the land surface model at every time step to modify the land surface processes due to $O_3$ damage, and the land surface variables simulated by land surface model to be dynamically passed back onto the atmospheric components, thus allowing immediate, two-way feedback effects onto meteorological fields, $O_3$ and other atmospheric chemical constituents, which is the major benefit of the two-way coupling (Section 2.2, L206-L212) in both terms of representing the real-world interactions more realistically and allowing coevolution of vegetation-atmosphere in future predictions.

Previous studies using offline (L94-L99) and two-way coupled models (L101-L116) have been introduced in the introduction section and in Table S15. The comparison between our results with those from offline models are now also discussed in L374-L386, L535-541.The downsides of the two-way coupled model of this study are now discussed in L544-L572.